# Unveiling Complex Collective Behaviors from Simple Rewards

## Abstract

Recent studies have shown that multiple agents trained through reinforcement learning can surprisingly exhibit swarm behaviors from simple rewards, without any rewards specifically encouraging aggregation. Explaining how complex collective behavior emerges from these simple rewards is an intriguing research problem, but the underlying process remains a black box up to now. This paper aims to reveal the hidden rules in this process. Specifically, we discovered that the reason agents are able to develop complex behaviors from simple rewards is that they implicitly learn the geometric fields of the environment and utilize these structures as desired targets for coordinated movement. This finding is supported by two distinct tasks: a competitive predator-prey pursuit-evasion and a cooperative multi-robot shape assembly. 1) In the competitive environment, prey agents surprisingly converge toward the boundary of the predators' Voronoi diagram, demonstrating that they are able to spontaneously learn Voronoi diagrams without any guided rewards. To gain the above insights, we propose a two-stage EEC (*Ego-observation → Ego-behavior → Collective-behavior*) explanatory framework. This includes a novel analytical tool called the Agent Response Map (ARM), which reveals agents' decision-making patterns across space and identifies regions of aggregation and avoidance. 2) The proposed method is extended to a more realistic and challenging cooperative robot-swarm task: shape assembly, to validate its generality and practical utility. The insights and tools presented in this paper may provide a new perspective on the connection between AI-driven multi-agent systems and real-world biological systems.

## 1 Introduction

Collective behaviors (e.g., starling flocks, fish schools, and sheep herds) are complex and fascinating phenomenon in nature (Sumpter, 2010), attracting widespread attention across various fields. A prevailing view is that survival pressure plays a central role in driving animals to form groups, as collective behavior can increase individual survival chances in the face of predation or environmental risks (Beauchamp, 2013). Yet rigorously quantifying and explaining such phenomena remains challenging. Various traditional models have been proposed to explain these collective behaviors, constructed from hand-designed rules (Vicsek et al., 1995; Reynolds, 1987; Couzin et al., 2002). While these models can replicate certain emergent patterns, their simplified interaction mechanisms fall short of capturing the complexity of real biological systems.

Recently, reinforcement learning (RL) has been used to simulate adaptive strategies in biological organisms (Durve et al., 2020; Monter et al., 2023), with multi-agent reinforcement learning (MARL) enabling simulations of complex agent interactions and optimizing collective behavior via rewards. For example, Durve et al. (2020) penalizes agents that "lose neighbors," while Monter et al. (2023) encourages prey to gather by reducing dangerous areas. However, these methods often rely on explicit hand-crafted rewards and therefore provide limited fundamental insights into why natural organisms would follow such designed rules.

In contrast, a recent study (Li et al., 2023) adopts a survival-pressure–based reward function and removes other hand-crafted reward terms within a predator–prey coevolution framework. It demonstrates the natural emergence of collective behaviors under simple incentives. This naturally raises the question: how can complex collective behaviors emerge from such simple rewards? Clarifying

this mechanism provides a mechanistic account of emergence and informs principled reward design for collective intelligence. However, existing interpretability work (Bush et al., 2025; Delfosse et al., 2024) struggles to systematically explain why local behavior at the individual level can give rise to swarm intelligence.

This paper aims to address this question. We mechanistically analyze the causes of emergent collective behavior, and surprisingly discover that agents implicitly learn latent risk fields and geometric invariants of the environment, exploiting these structures as desired targets for coordinated motion. It is interesting to see that agents converge toward the boundary of the Voronoi diagram (Okabe et al., 2009). A Voronoi diagram partitions space by proximity to reference points, with boundaries representing locations equidistant to multiple points. Voronoi diagrams are widely used across domains such as robotics, computer graphics, and spatial analysis, yet they typically rely on hand-crafted design rather than emerging spontaneously. We further show that this boundary functions as a line attractor—a line of stable points in state space, thereby providing insight into why simple rewards can give rise to complex collective behavior.

These findings are enabled by two-stage EEC (*Ego-observation → Ego-behavior → Collective-behavior*) explanatory framework, that traces how egocentric inputs drive individual actions and how those actions scale to swarm patterns: 1) *Ego-observation → Ego-behavior*, where key influential observation features are identified through SHAP (SHapley Additive exPlanations) (Lundberg & Lee, 2017) to quantify their contributions to the learned policy, followed by ablation experiments to validate the causal influence. This allows us to isolate the features most critical to the RL network's decision-making; 2) *Ego-behavior → Collective-behavior*, where we introduce the Agent Response Map (ARM), a novel explanation tool that reveals agents' decisions under different observations, and explains how ego-behavior yields swarm behavior by exposing common geometric safety fields shared across observations.

To validate effectiveness, we compare the SHAP results against attribution baselines: saliency map (Simonyan et al., 2013) and integrated gradients (Sundararajan et al., 2017), observing consistent attributions. SHAP demonstrates the advantage of providing additive, signed, and comparable attributions across heterogeneous observation features. On the other hand, ARM is, to our knowledge, the first tool to reveal how multi-agent policies condition on observations at a global level. So no direct baseline exists. Instead, we validate ARM using three qualitative criteria: 1) Verify that the boundary of the predators' Voronoi diagram forms a line attractor by two different approaches: The first is manipulating predator positions to alter the Voronoi boundary geometry. The second is evaluating prey convergence under impulse perturbations against three baseline collective models. 2) Predictive validation: We utilize ARM-identified features to predict agent aggregation in actual rollouts, thereby quantifying causal influence rather than correlation. 3) Sensitivity analysis: We quantify changes in performance under variations in agent motion parameters.

To further assess the generality of our framework, we study a more realistic robot-swarm control task: shape assembly (Sun et al., 2023; Zhu et al., 2025). Unlike the pursuit–evasion setting where there is no global contraints, shape-assembly tasks require agents must fill a global target shape while maintaining spacing, avoiding collisions, and preserving a uniform formation. We apply the proposed method within MARL pipeline to this setting. These results suggest that understanding the mechanisms behind emergent collective behavior can provide new insights for MARL and inform principled design of robust swarm-control policies.

## 2 RELATED WORK

### 2.1 INTERPRETABLE REINFORCEMENT LEARNING

Interpretability research in reinforcement learning (RL) can be broadly classified into three categories (Milani et al., 2024): Feature Importance (FI), Learning Process (LP), and Policy-Level (PL) analysis.

1) FI methods focus on explaining agent actions by identifying the observation features that most influence decision-making at each time step. Common techniques include SHAP values (Lundberg & Lee, 2017), Integrated Gradients (Sundararajan et al., 2017), and saliency maps (Atrey et al., 2019; Greydanus et al., 2018; Chakraborti et al., 2020). Although FI methods have been applied to analyze agent behaviors in Multi-Agent Reinforcement Learning (MARL) (Heuillet et al., 2022), they

are typically limited to single-step decisions and struggle to capture long-term behavioral patterns. 2) LP methods aim to understand how the training process shapes agent behavior by identifying influential training samples or critical states (Gottesman et al., 2020) (Gottesman et al., 2020; Guo et al., 2021; Cheng et al., 2023). However, LP approaches generally do not explain action selection given a specific observation. 3) PL methods summarize long-term behavior through abstraction or representative trajectories (Topin & Veloso, 2019; Amir et al., 2019; Boggess et al., 2022). However, the high computational complexity restricts them to small-scale discrete environments, rendering them unsuitable for large-scale multi-agent scenarios. Another class of PL methods is to replace black-box networks with white-box models. Approaches include extracting editable tree programs (Kohler et al., 2024), using concept bottlenecks (Delfosse et al., 2024), or employing neuro-symbolic frameworks that integrate neural networks with logic reasoning (Luo et al., 2024; Marton et al., 2025; Shindo et al., 2025).

However, these methods primarily translate policies into rules without explaining the mechanistic origin of *why* such behaviors evolve. Furthermore, they focus largely on single-agent settings, failing to capture the complex emergent behaviors inherent to multi-agent systems.

## 2.2 COLLECTIVE BEHAVIOR GENERATION VIA MARL

Multi-Agent Reinforcement Learning (MARL) serves as a powerful framework for modeling natural collective behaviors. Studies have demonstrated emergent flocking and foraging (Hahn et al., 2019; 2020b), characterizing swarm formation as a sub-optimal Nash equilibrium or social dilemma (Hahn et al., 2020a). Coordinated herding (Ritz et al., 2021) and target-directed swimming (Tovey et al., 2024) have also been achieved, while scalability has been enhanced through local communication methods (Hüttenrauch et al., 2019). Notably, Heuthe et al. (2024) recently applied MARL to control 200 laser-actuated microrobots. By introducing a counterfactual reward scheme, the swarm learned to cooperatively transport cargo, mirroring ant colony dynamics.

However, these approaches typically rely on reward functions that explicitly encourage clustering or coordination. Such hand-crafted incentives risk embedding human heuristics, potentially diverging from the fundamental mechanisms driving natural swarm emergence.

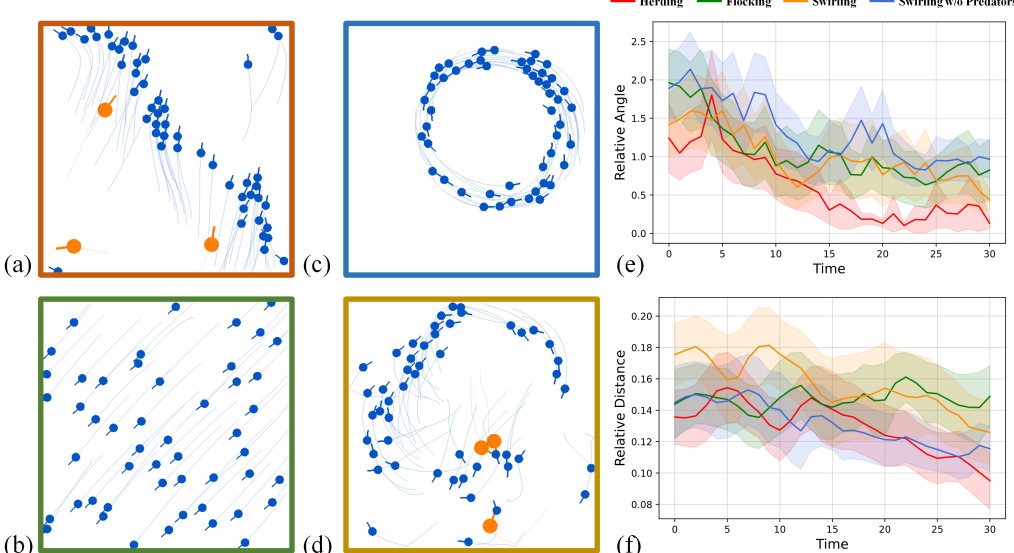

Figure 1: Agents are categorized into predators and prey, where orange circles represent predators and blue circles represent prey. Each agent is equipped with a short line segment indicating its orientation. (a) Herding. (b) Flocking. (c) Swirling without predators. (d) Swirling in the presence of predators. (e) Change in relative orientation angle between each prey and its nearest prey. The shaded area indicates a 95% confidence interval. (f) Change in relative distance over time.

## 3 PROBLEM SETUP

The predator-prey environment is a continuous space featuring two types of boundary conditions. The first is a confined square area, where agents cannot cross the boundaries modeled as walls with a specified contact stiffness, as commonly used in prior studies (Mordatch & Abbeel, 2018; Lowe et al., 2017; Heins et al., 2024; Huang et al., 2024). The second is a periodic boundary condition, where agents reappear on the opposite side after exiting the environment, forming a toroidal space. Periodic boundaries are widely employed in swarm modeling (Vicsek et al., 1995; Durve et al., 2020) to approximate large or infinite domains.

After MARL training, prey exhibit distinct collective behaviors across scenarios. Figure 1 (a)-(d) presents four scenarios where collective behaviors emerge. Specifically, (a) and (b) correspond to periodic boundary conditions, while (c) and (d) depict confined environments. In Figure 1(a), prey learned herding behavior in the presence of predators, characterized by two key features: First, the prey's orientations transition from random to aligned under predator influence. Second, their relative distances progressively decrease, as shown in Figure 1(e) and (f). In contrast, Figure 1(b) shows flocking behavior in the absence of predators, where prey maintain orientation alignment but without a further decrease in distance. Figure 1 (c) and (d) demonstrate the emergence of swirling behavior in the confined environment. In (c), without predators, prey spontaneously form circular motion patterns. In (d), with predators present, prey similarly form swirling patterns while simultaneously avoiding predators. During this process, prey also align their orientations and reduce their relative distances, shown in Figure 1(e) and (f).

The dynamics of agent motion can be described as:

$$
\begin{aligned}
\theta(t+1) &= \theta(t) + a_{\mathrm{R}}\Delta t \\
v(t+1) &= v(t) + (a_{\mathrm{F}}h + f_{\mathrm{d}} + f_{\mathrm{a}})\,\Delta t/m_i \\
x(t+1) &= x(t) + v(t)\Delta t
\end{aligned}
\tag{1}
$$

where $\Delta t$ represents the time step, $\theta \in (-\pi, \pi] \subset \mathbb{R}$ is the orientation angle. $a_R \in \mathbb{R}$ and $a_F \in \mathbb{R}$ are the two actions that an agent can perform, representing angular velocity and acceleration, respectively. $h = [\cos\theta, \sin\theta]^{\mathrm{T}} \in \mathbb{R}^2$ represents the agent's orientation. $f_{\mathrm{d}} \in \mathbb{R}^2$ and $f_{\mathrm{a}} \in \mathbb{R}^2$ are the drag force and elastic force caused by collisions between agents, respectively. $x \in \mathbb{R}^2$ represents the position of the agent, and $v \in \mathbb{R}^2$ represents the velocity, while $m_i \in \mathbb{R}^+$ represents the mass.

The observation vector for each agent in this environment is as follows:

$$
\begin{bmatrix}
\text{Agent's own position, velocity, and orientation} \\
\text{Relative position and orientation of each predator observed} \\
\text{Relative position and orientation of each prey observed}
\end{bmatrix}
\tag{2}
$$

Inspired by biological interaction constraints (e.g., panoramic vision in birds or lateral line sensing in fish (Liu et al., 2023; Ballerini et al., 2008)), the observation vector encodes the relative position and orientation of the nearest 6 prey and 6 predators within an omnidirectional field of view, ordered by proximity. This fixed-range, partially observable setting aligns with standard conventions in the MARL domain (Lin & Lee, 2024; Yu et al., 2022) and requires agents to actively prioritize spatially distributed risks. Zero-padding is applied when fewer neighbors are detected. Appendix A.2 confirms the robustness of emergent behaviors to alternative perception parameter settings.

Rewards are grounded in survival instincts: predators gain $r_{\mathrm{predator}} = +1$ and prey incur $r_{\mathrm{prey}} = -1$ upon physical collision. This reward mechanism operates independently of the emergent collective behavior and does not directly incentivize grouping. Despite using only this simple reward function, complex collective behaviors emerge. Additionally, we introduce an energy-consumption penalty $-0.01|a_F| - 0.1|a_R|$, to better reflect the cost of movement in nature. Robustness to alternative reward settings and population sizes is demonstrated in Appendices A.2 and A.3.

Collective behaviors emerge under various actor-critic MARL frameworks, such as MAPPO (Kölle et al., 2024; Yu et al., 2022) and MADDPG (Li et al., 2023; Lowe et al., 2017). Both actor and critic networks employ 3-layer feed-forward architectures (64 units, ReLU). The actor outputs a 2-D action vector, while the critic estimates a scalar value. Full training details are provided in Appendix A.1.

Emergent coordination thus arises from shared pressures and implicit biases rather than direct supervision. Understanding this mechanism requires moving beyond behavioral observation to examine how RL networks interpret and respond to input features.

## 4  EEC EXPLANATION FRAMEWORK

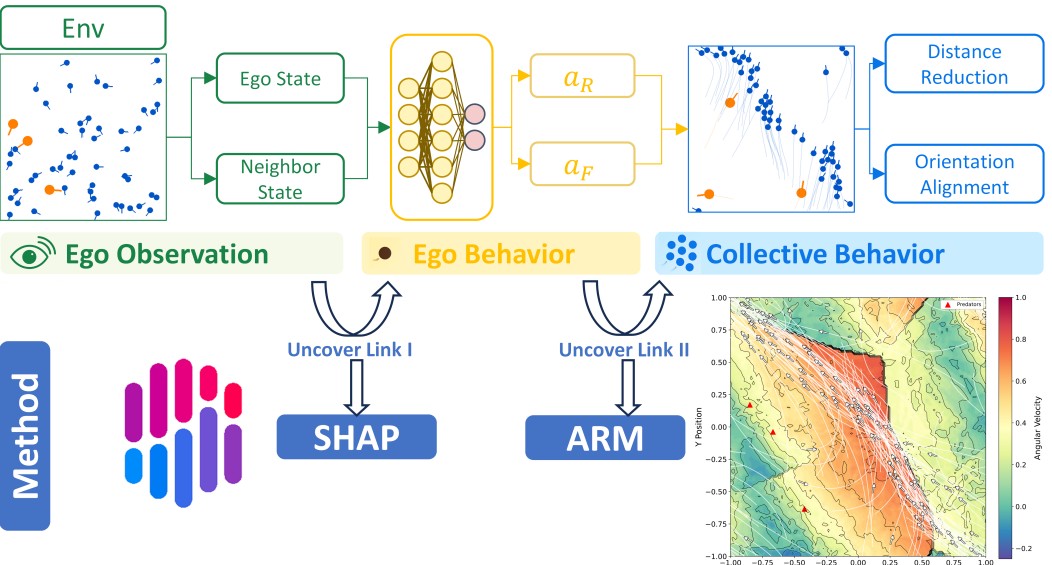

Figure 2: The proposed EEC explanation scheme comprises two components: 1) *Ego-observation → Ego-behavior*, which reveals how ego-behavior depends on ego-observation by analyzing the influence of individual features in the observation vector on an agent's decisions and identifying the most influential features; 2) *Ego-behavior → Collective-behavior*, which uses ARM to analyze how these feature-driven behaviors in the environment give rise to collective phenomena. An example of the ARM of $a_R$ under the periodic boundary condition is shown in bottom right, where red triangles denote predators which are currently static, white lines depict real prey trajectories, and arrows indicate movement direction. Black contour lines show the spatial level sets of $a_R$.

We aim to explain why simple rewards alone can induce the diverse collective behaviors. The proposed EEC explanatory framework aims to uncover the mechanisms behind emergent swarming by exposing two hidden links from ego observation to global coordination, as illustrated in Figure 2.

To decode the ego-observation to ego-behavior mapping (*Ego-observation → Ego-behavior*), we employ SHAP (Lundberg & Lee, 2017) to quantify feature influence. To distinguish causal necessity from correlation, we conduct ablation studies by retraining policies with masked features; features whose removal collapses collective behavior are deemed critical. These findings are corroborated by consistent results from baseline attribution methods, including saliency maps and integrated gradients.

To reveal how ego-behavior scales to collective behavior, we introduce ARM, which explains policy responses across space and explains how local decisions organize into swarm patterns. This is achieved by introducing a virtual prey agent that is invisible to real agents and therefore does not influence their behavior. The virtual agent acts as a probe to query the prey's response at different spatial locations at the current timestep, which can avoid the non-stationarity issue caused by mutual interference among agents in multi-agent systems. In this way, we can construct separate ARMs to visualize how key features shape prey movement, leading to orientation alignment and distance reduction—two hallmarks of collective behavior.

An example of ARM is shown in Figure 2. The color of each pixel represents the policy output for the virtual prey's angular velocity $a_R$ when it is placed at that location and its induced observation is fed into the policy network. We make three observations: 1) We observe that prey, initially scattered, gradually converge to a fixed region and flow coherently. This region is automatically highlighted by multiple black polylines through ARM. In fact, this region corresponds to the boundary of the predators' Voronoi diagram (Okabe et al., 2009), further indicating that it forms a line attractor—a line of stable points in state space: When a prey leaves this region, it quickly converges back; 2) The colors on the two sides of the Voronoi boundary differ, indicating distinct response regimes; 3) The color distribution depends on predator locations, i.e., the prey adjusts $a_R$ according to its relative

position to the predators. In Section 5.2, we provide concrete evidence and explanations for these three findings.

In the following, we aim to analyze the mechanisms underlying the emergence of herding behavior shown in Figure 1(a), while (b)-(d) shown in Appendix D and E.

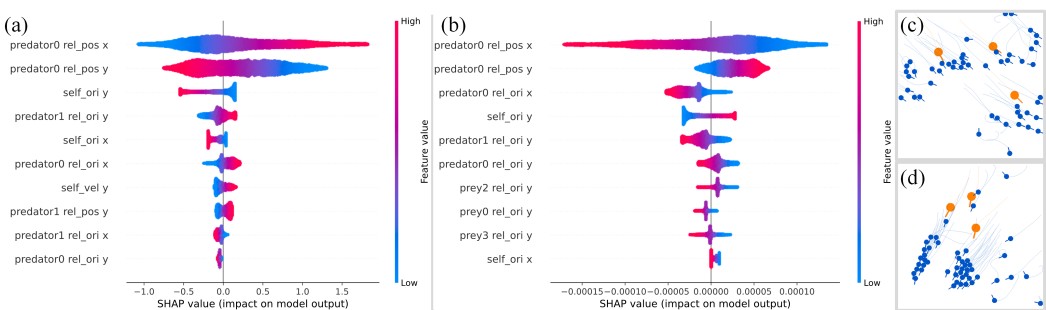

Figure 3: (a) SHAP analysis shows the top 10 observation features influencing $a_R$. The $x$-axis represents SHAP value, while the $y$-axis lists the corresponding features. Positive SHAP values indicate positive contributions to the action, negative values indicate inhibitory effects, and larger magnitudes denote stronger influence. The color bar reflects the feature value, with red for high values and blue for low values. (b) SHAP analysis of $a_F$. (c) Ablation: Removing orientation features. (d) Ablation: Fixing acceleration at maximum.

## 5 ANALYSIS OF COLLECTIVE BEHAVIOR

### 5.1 FROM EGO-OBSERVATION TO EGO-BEHAVIOR

To assess feature contributions to the policy network, we employ SHAP (Lundberg & Lee, 2017). Unlike standard i.i.d. applications, we apply SHAP to a multi-agent, partially observable, continuous-control setting, attributing egocentric features (relative positions/orientations) to policy outputs. The analysis uses observation data collected from 50 prey agents over 500 time steps, with 100 random background samples.

**1) Analysis of Observation Features Influencing Action $a_R$:** It is shown that the nearest predator's relative position has the strongest effect on $a_R$, demonstrated in Figure 3(a). This suggests prey adjust orientation primarily based on predator proximity.

Despite an isotropic environment, the $x$ and $y$ components yield different SHAP values. This likely stems from the non-i.i.d. nature of RL data inducing directional bias and the context-dependent nature of SHAP attributions (e.g., identical ranges require different turns depending on bearing). These limitations motivate our ARM method (Sec. 5.2).

To verify that the feature importance identified by SHAP reflects causality rather than only correlation, we conducted causal feature ablation studies by retraining the policy with specific features masked and tested on 20 different random seeds. First, removing neighbor orientation features did not hinder alignment or collective behavior (Figure 3(c)). Conversely, removing relative position features completely eliminated herding (Appendix B.1). This causal intervention confirms that relative position, not orientation, is the critical feature for collective coordination.

**2) Analysis of Observation Features Influencing Action $a_F$:** Counterintuitively, $a_F$ shows weak correlation with observation features (Figure 3(b)). Even the predator's position, despite being the top contributor, has a negligible SHAP value, indicating limited environmental influence on $a_F$.

To validate the causal role of acceleration, we fixed $a_F$ to its maximum value during execution. As shown in Figure 3(d), prey still formed compact groups, confirming that herding is not driven by $a_F$. Consistency checks using saliency maps (Simonyan et al., 2013) and integrated gradients (Sundararajan et al., 2017) align with these findings (Appendix B.2).

**3) Summation:** Taken together, the SHAP and ablation analyses show that the relative position of the nearest predator is the dominant factor influencing $a_R$, while $a_F$ exhibits limited sensitivity to observations. We thus focus next on how predator position shapes prey decisions via $a_R$.

## 5.2 FROM EGO-BEHAVIOR TO COLLECTIVE BEHAVIOR

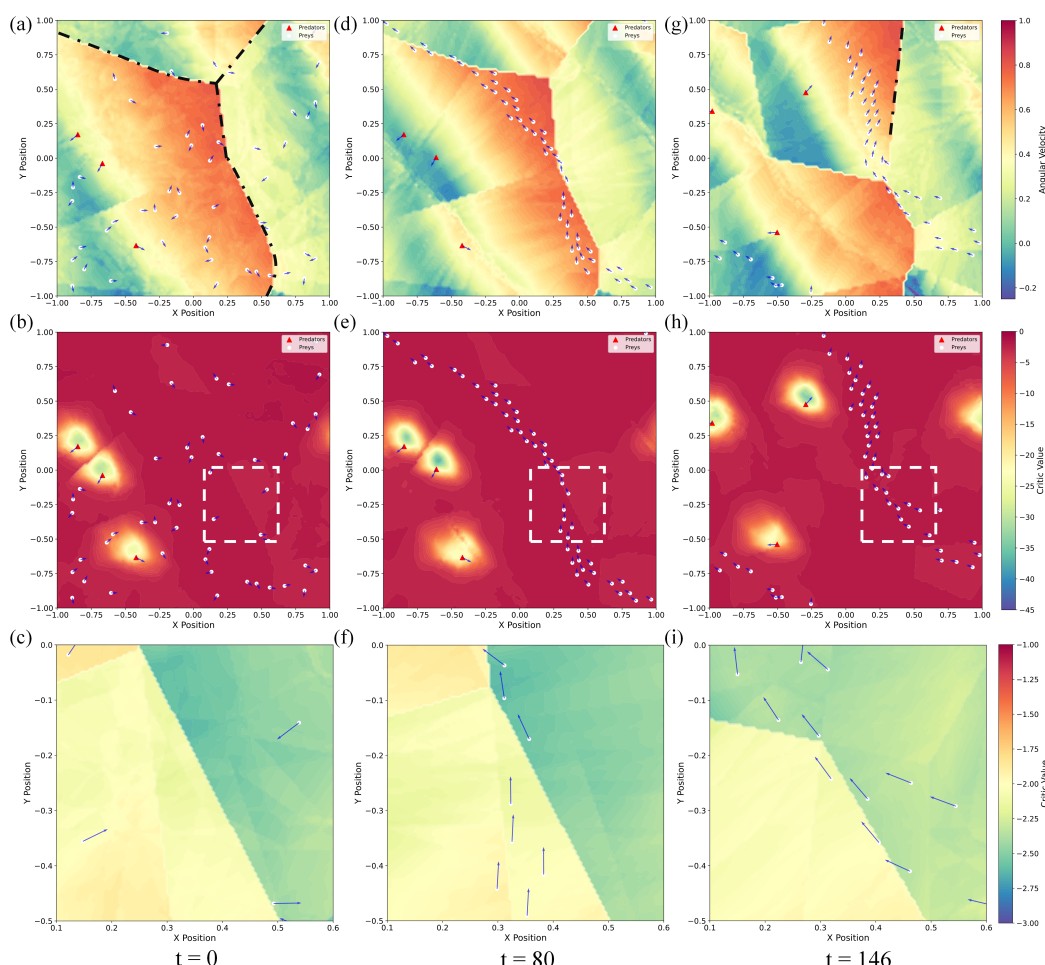

Figure 4: ARM for herding behavior: Each column corresponds to a time step. The first row shows the ARM of $a_R$, the second row shows the ARM of the critic network, the third row zooms into the white box region from the second row. Color intensity indicates the output magnitude. Red triangles denote predators, white circles represent prey, and arrows indicate orientation.

In this section, we introduce ARM to show how ego-behavior gives rise to collective behavior, which focuses on two salient signatures: orientation alignment and the decrease in relative distance.

### 5.2.1 EXPLANATION FOR ORIENTATION ALIGNMENT

Due to continuous interactions among agents, the environment evolves dynamically, making consistent behavioral patterns hard to isolate. To control for this, we keep the predator stationary from $t = 0$ to $t = 100$, then let it act under its learned policy, and fix it again for $t \geq 150$. This setup allows us to isolate and analyze the prey's behavioral responses under stable and transitional conditions.

To visualize how the policy and value functions vary across space, we construct the ARM: A virtual prey agent is introduced with zero velocity and an orientation angle of $\theta = 0$ (i.e., $h = [1, 0]^T$, pointing along the positive x-axis). This virtual agent records the observed data across the environment space, and feeds them into the trained policy and critic networks. The resulting policy output $a_R$ and

critic value are then mapped onto the environment space to form ARM visualizations. Note that the choice of orientation and velocity is arbitrary and serves as a simple default. In Appendix C.1, we evaluate the robustness of our findings by testing the impact of various virtual agent parameters on ARM results, including orientation angle, velocity, sampling resolution, agent population size, and noise interference.

It is shown that the prey's objective is to reach and move along the boundary of the predators' Voronoi diagram, which is equidistant from the predator along two opposing directions, demonstrated in Figure 4. It presents ARM results at three representative times: initialization $t = 0$, stationary predators $t = 80$, and predator pursuit $t = 146$. These maps show how decisions and values vary spatially with predator presence and prey distribution. At $t = 0$, prey are randomly distributed. As shown in Figure 4(a), the prey's angular velocity depends on predator position and exhibits discontinuities along the dashed black lines. These lines define a "safety zone", a region equidistant from the predator along two opposing directions. This region corresponds to the boundary of the predator's Voronoi diagram. The predator may approach from either side due to the periodic boundary condition described in Section 3. Prey on opposite sides thus adjust their orientations differently. The corresponding critic values in Figure 4(b) increase with distance from the predator, indicating lower risk and higher expected survival in farther regions. A zoomed-in view in Figure 4(c) reveals a discontinuity in critic values across the safety zone, consistent with abrupt changes in the perceived location of the nearest predator.

At $t = 80$, the predator remains stationary and prey converge into a narrow band aligned with the safety zone (Figure 4(d)). The ARM critic in Figure 4(e)–(f) peaks on the safety zone, indicating migration toward high-value regions to reduce risk—a learned self-protection strategy from the critic's spatial values.

Based on the analysis above, the prey's objective is to reach and move along the safety zone—the boundary of the predators' Voronoi diagram—where critic values are high. With the predator fixed, the zone's position and shape are fixed, enabling orderly motion along it.

At $t = 146$, the predator begins to chase. Figure 4(g) shows the shifted safety zone (dashed lines), prompting orientation updates. Prey near the zone share similar relative positions and thus adopt similar orientations, explaining alignment under pursuit.

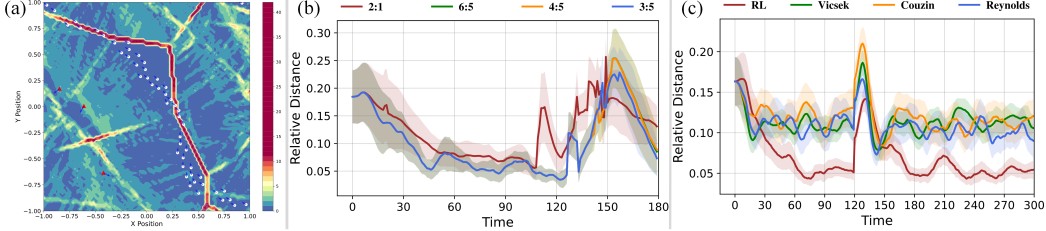

Figure 5: (a) Spatial Gradient Magnitude of ARM. Color intensity indicates the magnitude of gradient. (b) The prey's relative distance to the boundary of the predators' Voronoi diagram under four different $v_{\text{predator}} : v_{\text{prey}}$ speed ratios. The shaded area indicates a $95\%$ confidence interval. (c) The RL results compared with three rule-based baselines (Vicsek et al., 1995; Reynolds, 1987; Couzin et al., 2002).

We validate the ARM results using three quantitative approaches. First, we introduce the Spatial Gradient Magnitude (SGM) of ARM to quantify discontinuities by calculating the gradient magnitude of ARM outputs with respect to spatial position. Higher values indicate sharper action transitions. As shown in Figure 5(a), discontinuity peaks near the Voronoi boundaries where the magnitude reaches up to $40$, whereas it remains below $5$ in the vast majority of the environment. This result is consistent with our ARM analysis. Furthermore, we verified the robustness of the SGM method across different random seeds (see Appendix C.4).

Second, we utilize the SGM to predict prey agent aggregation behavior using two performance metrics: the concentration ratio (the ratio of the mean gradient at agent locations to the global mean gradient) and the percentile rank (the ranking of agent-location gradients within the global distribution). Averaged over 20 random seeds, the gradient at agent locations is $1.52$x higher than the global baseline, and agents consistently aggregate in the top $10.2\%$ of high-gradient regions, indicating high predictive precision.

Third, we verify that the Voronoi boundary forms a line attractor by tracking the relative distance between prey and the boundary (Figure 5(b, c)). Predator movement shifts the Voronoi boundaries. In (b), we evaluate prey reconvergence under such movement across four speed ratios using two metrics: steady-state distance and convergence time. Initially large, the distance decreases to a steady-state distance near $0.05$. At $t = 100$, predator motion increases the distance, causing non-smooth jumps due to discrete boundary shifts. Once predators stop ($t \geq 150$), the distance recovers to the steady-state level within approximately 30 seconds. Notably, the $2:1$ speed ratio induces sharper deviations and slower recovery. In (c), we compare our RL model against three swarm baseline models (Vicsek et al., 1995; Reynolds, 1987; Couzin et al., 2002) with static predators. The RL model achieves a steady-state distance of $0.05$, outperforming the baselines ($\approx 0.10$). Upon applying an impulse perturbation at $t = 120$, the deviation distance of the RL model remains below $0.15$ and rapidly reconverges, whereas baseline deviations all exceed $0.15$. This stability analysis confirms the boundary's attractor property and suggests the RL-learned policy approximates an optimal strategy.

We further validate the robustness of this phenomenon in Appendix C.2, where ARM results under different random seeds consistently exhibit similar spatial patterns, confirming that this is a general and repeatable mechanism underlying herding behavior.

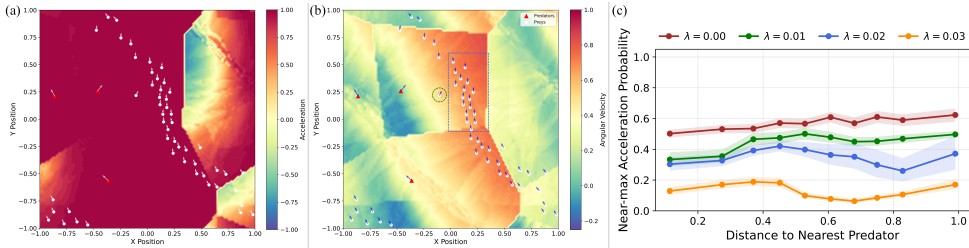

Figure 6: ARM illustrating the decrease in relative distance: (a) ARM of $a_{\mathrm{F}}$. (b) ARM showing that prey at different distances from the predator exhibit distinct orientations. (c) Quantile-binned probability of near-max acceleration as a function of predator distance, with $95\%$ confidence interval.

### 5.2.2 EXPLANATION FOR DECREASE IN RELATIVE DISTANCE

Figure 6(a) presents the ARM of $a_{\mathrm{F}}$ at $t = 140$ (setting from Sec. 5.2.1). Results indicate that prey move at near-maximal acceleration ($[-1, +1]$) across most of the environment; nevertheless, their relative distances continue to shrink. This aligns with Figure 3(c-d), confirming that distance reduction is not primarily driven by acceleration magnitude.

We quantify the relationship between prey acceleration and relative distance to the nearest predator using quantile binning (K=10) on predator distance to obtain equal-frequency bins. Within each bin, we estimate the probability of near-max acceleration ($a_{\mathrm{F}} \geq \tau$) and reported $95\%$ confidence interval. Here, $\tau = 0.95$ serves as a threshold for near-maximum acceleration. We also vary the energy-consumption penalty in the RL reward $-\lambda |a_{\mathrm{F}}|$, and examine the trained outcomes. Results are shown in Figure 6(c). We observe that: (1) the probability of maintaining near-max acceleration is relatively flat, remaining approximately consistent across distances; and (2) stronger acceleration penalties reduce the probability of near-max acceleration, i.e., as $\lambda$ increases, the probability decreases.

Controlling for acceleration, we treat angular velocity $a_{\mathrm{R}}$ as the explanatory variable. In Figure 6(b), prey farther from the safety zone (green circle) orient differently from those near it (blue box): the former exhibit an $x$-directed component toward the group. Consequently, even under equal acceleration, farther prey have a larger $x$-velocity component, move toward the group center along $x$-axis, and reduce relative distances. Appendix C.3 provides additional results on how the critic and policy ARMs evolve over training.

## 6 EXTENSION: MULTI-ROBOT SHAPE ASSEMBLY

The task of multi-robot shape assembly comprises two steps: 1) We specify the desired target shape; 2) Robots use this shape as a goal and, through continuous local inter-

actions with neighbors during motion, uniformly fill the shape, as depicted in Figure 7. The target region has a connected shape and is discretized into a grid of $n_{cell}$ cells. Each robot's action is a two-dimensional vector with components along the $x$ and $y$ axes, This vector indicates the active force $f_a$. The passive force, $f_b$ is an elastic force following Hooke's Law. The observation vector of each agent consists of four parts: The first part is the robot's own state, the second is the relative state of its neighbors, the third is the relative position of the target cell, and the fourth is the relative positions of unoccupied/observed cells within $r_{sense}$. The maximum number of neighbors and observed cells are denoted as $n_{hn}$ and $n_{hc}$, respectively. Hence, each robot's observation is a $(6+4n_{hn}+2n_{hc})$-dimensional vector. The reward function design is

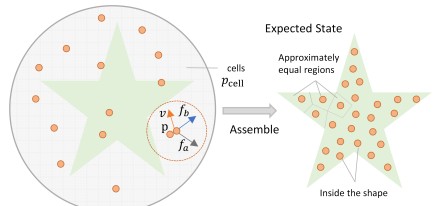

Figure 7: Shape assembly environment description. Agents are required to navigate to designated regions and self-organize into a user-specified target shape. In our experiments, the target shape is a five-pointed star.

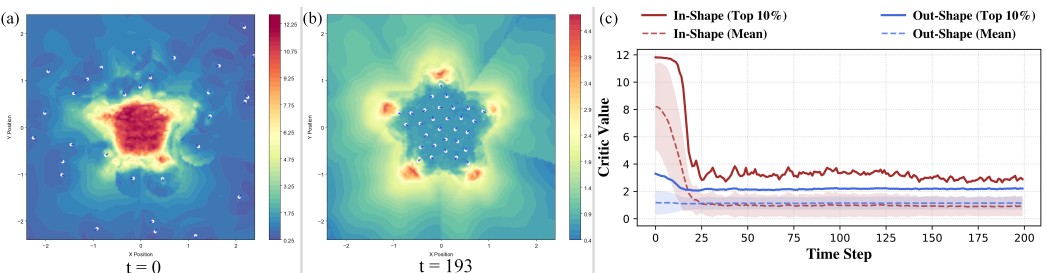

Figure 8: (a)-(b) ARM of critic network at different time steps. (c) Comparison of the overall mean and the top-10% mean of critic values within In-Shape versus Out-Shape regions.

considered to satisfy three conditions. If all conditions are met, the reward is 1; otherwise, it is 0: (1) The robot is within the shape. (2) The robot avoids collisions with neighbors. (3) Encourage robots to explore unoccupied area, specifically: $\left| \sum_{k \in \mathcal{C}_i} \rho_k p_k / \sum_{k \in \mathcal{C}_i} \rho_k - p_i \right| \leq \delta, i \in \mathcal{A}$, where $\rho_k = 0.5(1 + \cos \pi \|p_k - p_i\|/r_{sense})$, $\mathcal{C}_i$ is the sets of neighbors/observed cells of robot and $\delta = 0.05$.

From Figure 8(a), (b), we observe that initially, the unoccupied target center exhibits high critic values, drawing agents inward. As the center fills, the value peak shifts to the boundary, promoting the exploration of unoccupied areas. This mirrors the design intuition in (Sun et al., 2023). We support this finding with quantitative analysis in Figure 8(c), comparing the overall mean and top 10% mean critic values inside versus outside the target shape. Initially, internal values far exceed external ones. As the task progresses, the internal mean value drops due to agent occupancy, while external values remain consistently low. The In-Shape Top 10% value remains much higher than the Out-Shape. This confirms that the Critic network successfully identifies the target interior as the region of highest global value. Metric stabilization around $t = 50$ indicates convergence to a stable geometric attractor. These results validate the effectiveness of ARM in identifying meaningful geometric invariants in task-driven cooperative MARL, demonstrating the generality of our method.

Additionally, we validated the framework on complex non-convex and multi-component shapes, assessing robustness against uncertainties like noise. Appendix F details these results.

## 7 CONCLUSION

This paper uses multi-agent reinforcement learning (MARL) to explore the hidden mechanisms of collective behavior. Agents trained with only simple rewards exhibit swarm behavior without aggregation incentives, because they implicitly learn geometric structure and use it as desired targets for coordinated motion. A two-stage EEC explanatory framework is proposed which includes a novel analytical tool called the Agent Response Map (ARM), and reveals agents' decision-making patterns across space and identifies regions of aggregation and avoidance. These findings offer new insights into collective behavior in MARL, and the proposed framework is applicable to broader multi-agent settings.

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
