# OpenReview forum: "Unveiling Complex Collective Behaviors from Simple Rewards"
_ICLR.cc/2026/Conference — Submitted to ICLR 2026_

### Official Review · Reviewer_NZCx · 2025-10-24

**Soundness:** 3
**Presentation:** 1
**Contribution:** 2
**Rating:** 2
**Confidence:** 4

**Summary:**

The authors introduce approaches to analyze multi-agent reinforcement learning experiments, referred to as EEC. These include Ego-observation -> Ego-behavior, followed by Ego-behavior -> Collective behavior. The first stage of this analysis is the application of SHAP to trained actor and critic networks. In the second stage, a so-called Agent Response Map (ARM) is constructed, showing how agents respond to different scenarios within a simulation in order to shed light on their collective behavior.

In the paper, the authors focus on a predator-prey model with the prey punished for being captured and the predators being rewarded for capturing. As input, the agents receive information about their position along with the relative position of the other agents in the simulation. During ablation studies, this information is limited. The authors find via SHAP analysis that relative position is the essential ingredient in producing collective behavior in the prey. Further, they identify that the prey appear to move towards the edges of the Voronoi diagram formed using the predator agents as nodes. An additional study is mentioned relating to a robotics task, but the results are not presented in the main paper.

The approach to analyzing the learned policy can be seen as an interesting result and contribution of the paper. The learned policy of the prey, essentially to maximize their distance from predators, is nice to see, but not necessarily of great surprise.

**Strengths:**

The paper tackles an important problem in reinforcement learning, namely, better understanding the emergent policy of agents. Further, the visual representation of the policy analysis makes the outcomes clear to readers. Further, by explaining the emergent policy of the agents, the authors shed light on a common RL problem.

**Weaknesses:**

There were several concerns raised while reading the paper. First, the concerns regarding the study will be addressed:

- A predator is mentioned in the abstract before any mention of the kinds of problems that were studied
- At times, it felt as though the prior work was not adequately addressed. Ignoring the fact that the section does not appear in the main paper, even in the appendix, and with the language used in the main text, there are works approaching this problem on very similar systems. E.g., https://arxiv.org/pdf/2110.01307, 10.1088/2632-2153/ad5f73 as two notable examples.
- In some cases, the analysis is convoluted. Discussing Voronoi diagrams is reasonable, but is the central point that the agents maintain a maximum distance from the predators? This is a far simpler explanation for what appears to be the same result.
- Due to the problem addressed, while interesting, the results are not particularly novel.
- Throughout the paper, as will be addressed below, critical details such as the RL algorithm, information on PBC, and other factors are simply not discussed. This makes understanding the results difficult and reproducibility all but impossible.

Outside of the study, it is also important to mention the following concerns:

- Moving related work to the appendix is not an appropriate way to reduce the size of a paper. Those rules are put in place for a reason. This also goes for removing the RL theory from the paper. It is never clearly stated what kind of RL is being studied. The authors mention actor and critic, thereby insinuating it, but no details are provided. This also goes for the lack of detail regarding the simulations and reproducibility. This is also done in the case of the final study in a robotic setting. Overall, the paper cannot be considered self-contained.

**Questions:**

- Is the “sight” 360 degrees?
- Is the position given to the agents global? This would contradict the statement “the agents must coordinate their motion using only local information.”
- Is the ARM computed for fixed predator positions (This is explained later to be true, but needs to be clarified before Figure 2). In Figure 2, are the authors using PBC or not? Judging by the trajectories, it seems like they are. This changes the expected outcome and is thus critical information. Again, I believe this is mentioned later in the paper, but if the authors wish to include and mention these results in the figure, this information should be introduced at that point.
- In the absence of relative position information, is it surprising that the agents cannot form groups?
- When saying, “equidistant from the predator along two opposing directions,” is it meant equidistant between two predators? Hence, along the Voronoi diagram formed using those predators as nodes? This would come down to the agents maximizing their distance from both predators, not exactly a surprising result, although admittedly interesting that the agents learned this form of geometry given the reward used.
- Can the authors explain why the angular velocity is so high along the Voronoi lines? I would have expected that the agents would stop turning in these regions and move relatively straight

**Details Of Ethics Concerns:**

The authors have removed substantial information, such as prior work, simulation details, and reinforcement learning algorithm/training information, from the main text to the appendix in order to meet the page limit. Further, this is stated in the introduction of the paper:

"A detailed review of related work is provided in Appendix A due to page limits and will be moved to
the main text if additional space becomes available during the discussion/rebuttal phase."

I find this to be a breach of ethics for the conference.

---

> ### Author Response · Authors · 2025-12-03
> **Response to Reviewer NZCx (Weakness 1 &2)**
>
> > **Weakness 1: A predator is mentioned in the abstract before any mention of the kinds of problems that were studied.**
>
> **Response 1**: Thanks for your comments.  We added a description of kinds of problems we studied in abstract, shown as
> "This finding is supported by two distinct tasks: a competitive predator-prey pursuit-evasion and a cooperative multi-robot shape assembly. 1) In the competitive environment, prey agents surprisingly converge toward the boundary of the predators’ Voronoi diagram, demonstrating that they are able to spontaneously learn Voronoi diagrams without any guided rewards. "
>
>
>
> > **Weakness 2: At times, it felt as though the prior work was not adequately addressed. Ignoring the fact that the section does not appear in the main paper, even in the appendix, and with the language used in the main text, there are works approaching this problem on very similar systems. E.g., https://arxiv.org/pdf/2110.01307, 10.1088/2632-2153/ad5f73 as two notable examples.**
>
> **Response 2**:
> Thank you for your advice. While these two papers are relevant to the field, their focus differs from our study.
>
> * The first paper primarily employs Shapley values to analyze individual agent behavior within MARL. However, Shapley value analysis is inherently a feature attribution method. While effective for identifying input importance, it **cannot explain long-term temporal dynamics or elucidate how individual behaviors coordinate to yield global swarm behaviors**.
> * The second paper utilizes RL to guide agents toward a specific target. Its reward function is explicitly designed to encourage agents to approach the destination. Consequently, the formation of a cluster in that context is a **direct and expected result of hand-crafted reward shaping**, rather than a spontaneous phenomenon emergent from simple survival pressure.
>
> We have revised the Related Work section to include and discuss these studies.

---

> ### Author Response · Authors · 2025-12-03
> **Response to Reviewer NZCx (Weakness 3)**
>
> > **Weakness 3: In some cases, the analysis is convoluted. Discussing Voronoi diagrams is reasonable, but is the central point that the agents maintain a maximum distance from the predators? This is a far simpler explanation for what appears to be the same result.**
>
> **Response 3**: Thanks for your question.  Actually, "maximizing predator–prey distance" is not the only strategy for evasion.
> 1. **Diversity of Evasion Strategies**: In **nature**, escape tactics often involve non-trivial turning angles and protean (unpredictable) movement patterns rather than movement aimed solely at maximizing distance [1, 2, 3]. Similarly, within **control theory and pursuit–evasion literature**, various alternative strategies exist. These include modeling the problem as a differential game solved via Hamilton–Jacobi–Isaacs variational inequalities [4], or constructing strategies based on isochrones and Apollonius circles [5].
> 2. **Limitations of Prior Methods**: However, these traditional approaches typically rely on **hand-crafted designs** and suffer from **significant computational burdens**, making them ill-suited for large-scale multi-agent systems. Moreover, they generally **assume global state knowledge of all agents, which is often unrealistic**. In contrast, we consider a partially observable setting where agents perceive only a limited number of nearest neighbors.
> 3. **Our Contribution**: This paper establishes a link between Voronoi-based evasion and evolutionary survival pressure. Our work provides **the first mechanistic explanation for emergent collective behavior in MARL**. The ARM analysis finds that agents learn the underlying Voronoi geometry solely through survival pressure. By comparing our RL results with three classic collective models (Vicsek et al., 1995; Reynolds, 1987; Couzin et al., 2002), we demonstrate that the **RL agents identify the Voronoi boundary as the optimal solution**. These findings imply that complex collective behaviors in nature may emerge as evolutionary responses to survival pressure, offering new inspiration for designing robust robot controllers. Finally, we demonstrated the robustness of our results by testing across various random seeds, parameters, and noise levels.
>
> **References**：
>
> [1] Domenici, P., Blagburn, J. M., & Bacon, J. P. (2011). Animal escapology I: theoretical issues and emerging trends in escape trajectories. Journal of Experimental Biology, 214(15), 2463-2473.
>
> [2] Richardson, G., Dickinson, P., Burman, O. H., & Pike, T. W. (2018). Unpredictable movement as an anti-predator strategy. Proceedings of the Royal Society B, 285(1885), 20181112.
>
> [3] Miller, G. F., & Cliff, D. (1994). Protean behavior in dynamic games: Arguments for the co-evolution of pursuit-evasion tactics. From animals to animats, 3, 411-420.
>
> [4] Garcia, E., Casbeer, D. W., Von Moll, A., & Pachter, M. (2020). Multiple pursuer multiple evader differential games. IEEE Transactions on Automatic Control, 66(5), 2345-2350.
>
> [5] Li, S., Wang, C., & Xie, G. (2022, June). Pursuit-evasion differential games of players with different speeds in spaces of different dimensions. In 2022 American Control Conference (ACC) (pp. 1299-1304). IEEE.

---

> ### Author Response · Authors · 2025-12-03
> **Response to Reviewer NZCx (Weakness 4)**
>
> > **Weakness 4: Due to the problem addressed, while interesting, the results are not particularly novel.**
>
> **Response 4**: Thanks for your comments. This paper establishes a link between Voronoi-based evasion and evolutionary survival pressure. Our work provides **the first mechanistic explanation for emergent collective behavior in MARL**. Our MARL model also demonstrates optimal performance, shown in  Figure 5.
>
> Actually, the Voronoi boundary represents the theoretically optimal solution for risk minimization:
>
> * **Biology (Selfish Herd)**: Extensive biological research models individual predation risk as a function of the Voronoi cell ("domain of danger") [1]. Moving toward the boundary of these domains is an evolutionary optimal strategy to shift risk to neighbors or maximize safety margins.
>
> * **Robotics & Control (Optimal Path)**: In multi-obstacle or multi-threat environments, the "safest" path is to maximize the distance to the nearest obstacle/threat, which corresponds to the Voronoi boundary  [2,3]. Therefore, by converging to the Voronoi boundary, the RL agents are essentially approximating the optimal "Maximin" strategy (maximizing the minimum distance to any predator).
>
> However, while Voronoi-based control algorithms exist,  they are typically **hand-crafted with human priors**, making agents to calculate Voronoi cells [4], assuming the agents know the geometry beforehand. We do not know if biological evolution actually "computes" this or approximates it via other means. Furthermore, it assumes agents possess global geometric knowledge, and the real-time calculation of Voronoi boundaries is computationally expensive, limiting scalability for large clusters.
>
> **Our Contribution**:
> 1. This paper **establishes a link between Voronoi-based evasion and evolutionary survival pressure**: Under a simple survival-pressure reward (without any geometric hints), Deep RL spontaneously discovers this optimal geometric structure. **This suggests that the Voronoi strategy is a natural attractor in the policy space for survival tasks**: Nature evolves this strategy because it is the most efficient way to survive, not because it was programmed to do so. These findings imply that complex collective behaviors in nature may be evolutionary responses to survival pressure and offer new inspiration for designing robust robot controllers.
>
> 2. Furthermore, our explanation method is not limited to identifying Voronoi boundaries in competitive pursuit-evasion tasks, it is **applicable to cooperative tasks, such as shape assembly**. Unlike pursuit-evasion under periodic boundaries, shape-assembly tasks **require agents to fill a global target shape** while maintaining uniform spacing and avoiding collisions. We observe that when the target shape is initially unoccupied, its center exhibits high critic values, drawing agents inward. Once the center becomes occupied, the value peak shifts to the boundary, promoting the exploration of unoccupied areas. This dynamic mirrors the design intuition in Sun et al. (2023) and supports the validity of our explanatory method.
>
> In conclusion, the proposed EEC framework and ARM are not confined to specific pursuit-evasion scenarios. They are **capable of analyzing diverse MARL settings ranging from competitive to cooperative tasks**, to uncover the distinct hidden geometric structures governing agent behavior.
>
>
> **References**
>
> [1] Hamilton, William D. "Geometry for the selfish herd." Journal of theoretical Biology 31.2 (1971): 295-311.
>
> [2] Choset, H., & Burdick, J. (2000). Sensor-based exploration: The hierarchical generalized voronoi graph. The International Journal of Robotics Research, 19(2), 96-125.
>
> [3] Bakolas, E., & Tsiotras, P. (2010). Optimal pursuit of moving targets using dynamic Voronoi diagrams. In 49th IEEE conference on decision and control (CDC) (pp. 7431-7436). IEEE.
>
> [4] Bakolas, E., & Tsiotras, P. (2010). The Zermelo–Voronoi diagram: A dynamic partition problem. Automatica, 46(12), 2059-2067.

---

> ### Author Response · Authors · 2025-12-03
> **Response to Reviewer NZCx (Weakness 5-6)**
>
> > **Weakness 5: Throughout the paper, as will be addressed below, critical details such as the RL algorithm, information on PBC, and other factors are simply not discussed. This makes understanding the results difficult and reproducibility all but impossible.**
>
> **Response 5**: Thank you for your comment. In the original submission, the periodic boundary conditions were defined in the first paragraph of the *Problem Setup* section, while implementation details were initially placed in the Appendix.
>
> In the revised manuscript, we have moved the **detailed RL training and network architecture specifications** to the Main Text (Page 4, Line 210). Additionally, we have included comprehensive specifications regarding the **optimizer, hardware infrastructure**, and **training duration** in Appendix A.1. These revisions ensure that the experimental results can be fully reproduced from scratch.
>
> Regarding reproducibility, we have **provided the source code** in the supplementary material and are committed to making the codebase fully open-source to facilitate future research.
>
> > **Weakness 6: Moving related work to the appendix is not an appropriate way to reduce the size of a paper. Those rules are put in place for a reason. This also goes for removing the RL theory from the paper. It is never clearly stated what kind of RL is being studied. The authors mention actor and critic, thereby insinuating it, but no details are provided. This also goes for the lack of detail regarding the simulations and reproducibility. This is also done in the case of the final study in a robotic setting. Overall, the paper cannot be considered self-contained.**
>
> **Response 6**: Thank you for the suggestion. Due to the additional space allowance during the rebuttal phase, we have integrated the Related Work section into the main paper (Page 2).
> Simultaneously, we have relocated the detailed RL training and network architecture specifications to the Main Text (Page 4, Line 210). We also added comprehensive details regarding the **optimizer, hardware infrastructure**, and **training duration**. These revisions collectively ensure that the experimental results can be fully reproduced from scratch.

---

> ### Author Response · Authors · 2025-12-03
> **Response to Reviewer NZCx (Question 1-3)**
>
> > **Question 1: Is the “sight” 360 degrees?**
>
> **Response 7**: Thanks for your question. The agents' perception is omnidirectional ($360^\circ$). This is inspired by **biological interaction**: In nature, collective coordination relies not only on vision but also on omnidirectional sensory modalities, such as hearing, olfaction, or the lateral line system in fish. A $360^\circ$ range serves as a generalized abstraction of these systems.
>
> While omnidirectional sensing provides broad coverage, it introduces the challenge of sensory filtering and prioritization. Unlike a limited view cone that naturally filters out rear distractions, a $360^\circ$ view forces the agent to actively learn which direction is most critical (e.g., distinguishing between an immediate threat behind and a neighbor in front). This **requires the RL policy to develop a more robust spatial attention mechanism** rather than relying on hard-coded blind spots.
>
> **Standard MARL Setting**: Omnidirectional sensing is a standard convention in continuous control MARL benchmarks (Lin & Lee, 2024; Yu et al., 2022) to focus the challenge on interaction logic.
>
> **Partial Observability**: Despite the $360^\circ$ field of view, the environment remains partially observable. Agents are strictly limited by a fixed sensing radius and a maximum neighbor count ($k=6$), preventing access to global state information.
>
> We have updated Section 3 (Problem Setup) to explicitly clarify this mechanism.
>
> > **Question 2: Is the position given to the agents global? This would contradict the statement “the agents must coordinate their motion using only local information.”**
>
> **Response 8**: Thanks for your question. The positional information provided to the agents comprises two components：the agent's own position, and neighbor agents position.
> - **Own position**: Agents indeed have access to their own position/velocity (Proprioceptive).
> - **Neighbor position**: Crucially, agents cannot access the global positions of others. They only perceive neighbors within a fixed range and limited count ($k=6$), obtaining only relative positions (Exteroceptive).
>
> To avoid ambiguity and ensure precision, we have removed this statement ("...using only local information") in the revised manuscript.
>
>
> > **Question 3: Is the ARM computed for fixed predator positions (This is explained later to be true, but needs to be clarified before Figure 2). In Figure 2, are the authors using PBC or not? Judging by the trajectories, it seems like they are. This changes the expected outcome and is thus critical information. Again, I believe this is mentioned later in the paper, but if the authors wish to include and mention these results in the figure, this information should be introduced at that point.**
>
> **Response 9**: Thanks for your question. We have modified the description of Figure 2, shown as "An example of the ARM  of $a_{\mathrm{R}}$ under the periodic boundary condition is shown in bottom right, where red triangles denote predators which are currently static".

---

> ### Author Response · Authors · 2025-12-03
> **Response to Reviewer NZCx (Question 4-6)**
>
> > **Question 4: In the absence of relative position information, is it surprising that the agents cannot form groups?**
>
> **Response 10**:
> Thank you for your question. The fact that prey fail to aggregate after removing relative information is indeed expected, not surprising. As noted in Line 314 of the revised paper, the purpose of this ablation experiment was **not to present this failure as a novel discovery**, **but to causally validate the conclusion from our SHAP analysis**: The relative position is the dominant factor influencing coordination (specifically the angular velocity, $a_R$).
> However, **the truly counter-intuitive finding is that the agent's acceleration magnitude ($a_F$) does not significantly vary with the distance to the predator**. In other words, prey successfully reduce their inter-agent distance and form clusters while maintaining a constant speed. This is supported by two key observations:
> 1. **SHAP Analysis**: In contrast to the massive impact of relative position on $a_R$, the impact of all observed features on acceleration ($a_F$) is negligible.
> 2. **Causal Ablation**: Even more surprisingly, when we fixed $a_F$ to its maximum value during the ablation experiment, the prey still formed compact clusters.
>
> This leads to the core mechanism detailed in Section 5.2: *How do prey shorten their mutual distance while maintaining (nearly) identical maximum speeds?* The answer lies in the orientation control ($a_R$). Agents at the periphery of the swarm adjust their heading slightly toward the swarm center. This generates an inward-directed velocity component, allowing the swarm to compress without decelerating.
>
> > **Question 5: When saying, "equidistant from the predator along two opposing directions," is it meant equidistant between two predators? Hence, along the Voronoi diagram formed using those predators as nodes? This would come down to the agents maximizing their distance from both predators, not exactly a surprising result, although admittedly interesting that the agents learned this form of geometry given the reward used.**
>
> **Response 11**: This question is similar to **Weakness 3** and  **Weakness 4**. We refer to **Response 3 &4** for details.
>
>
> > **Question 6: Can the authors explain why the angular velocity is so high along the Voronoi lines? I would have expected that the agents would stop turning in these regions and move relatively straight**
>
> **Response 12**:
> Thank you for your question. On opposite sides of the Voronoi boundary, there is a sharp contrast in angular velocity values (e.g., high positive velocity on one side versus low or negative velocity on the other). This abrupt transition creates a **discontinuity magnitude specifically at the boundary**. Physically, this phenomenon arises because prey on opposite sides must rotate in divergent directions to effectively evade the predator.
>
> While the specific magnitude of these angular velocities depends on the heading of the virtual probe, the underlying structural discontinuity remains constant. **Appendix Figure 8 presents results using different virtual prey headings**: While the generated angular velocities vary, every configuration successfully captures the common Voronoi structure. **This demonstrates the robustness of ARM to variations in heading parameters.**

---

### Official Review · Reviewer_HkEp · 2025-10-29

**Soundness:** 3
**Presentation:** 3
**Contribution:** 2
**Rating:** 4
**Confidence:** 3

**Summary:**

This paper investigates the question of how complex collective behaviors, such as herding and flocking, can emerge in multi-agent reinforcement learning (MARL) systems trained with only simple survival-based rewards. The authors propose a two-stage explanatory framework, EEC (Ego-observation → Ego-behavior → Collective behavior), to dissect this process. The core contributions are: 1) a novel visualization tool called the Agent Response Map (ARM), which probes and maps the learned policy across the environment space, and 2) the surprising discovery that prey agents learn to converge towards the boundaries of the predators' Voronoi diagram, which acts as a "line attractor" or a dynamic safety zone. The paper uses SHAP and ablation studies to identify critical observational features and validates its findings in a predator-prey environment, with a brief extension to a robot shape-assembly task.

**Strengths:**

1.  The paper tackles a fundamental question at the intersection of AI, biology, and complex systems: the emergence of collective intelligence from simple, local rules. Providing a mechanistic explanation for this phenomenon in a modern MARL context could be a useful enabler for analyzing future algorithms.

2.  The proposed Agent Response Map (ARM) is a highly intuitive tool for interpreting MARL policies. Visualizing the policy's output as a field over the state space provides a global, geometric understanding of agent decision-making that is hard to obtain from trajectory analysis alone. This is a methodological contribution to MARL interpretability.

3.  The discovery that agents implicitly learn and exploit the Voronoi diagram of predators is interesting. The framing of this boundary as a "line attractor" is a powerful and clear explanatory concept.

**Weaknesses:**

1.  The work is dependent on the periodic boundary conditions and simple dynamics, which create a situation where a predator can approach from two opposing directions. Further, predator-prey problems in 2D are quite the special case and may not be of huge generality. The line attractor property is not theoretically backed.

2.  The extension to the multi-robot shape assembly task in Section 5 feels disconnected from the main analysis. The section is very brief, with most details relegated to the appendix. It does not clarify whether an analogous geometric structure (like the Voronoi diagram) emerges or how the EEC framework provides unique insights in that context.

3.  The ARM is constructed using a "virtual probe" agent with zero velocity. While this is a clever way to isolate the policy's spatial response, a trained policy is conditioned on the full state, including velocity.

**Questions:**

1.  Could you elaborate on the role of the periodic boundary conditions in the emergence of the Voronoi diagram as an attractor? Have you run experiments in a large, un-walled environment or on other topologies? If so, does a similar geometric structure emerge, or do the agents adopt a different strategy?

2.  Did you experiment with using a probe agent with non-zero (e.g., average) velocity? How sensitive is the structure of the ARM to the probe agent's own state, and would this change the interpretation of the results?

3.  The paper shows agents converge to this solution, but could you speculate more on why the MARL algorithm finds this specific Voronoi-based strategy? Is it demonstrably the most optimal path, or is it simply always the case for the simple problem dynamics with continuous policies?

---

> ### Author Response · Authors · 2025-12-03
> **Response to Reviewer HkEp (Weakness 1)**
>
> > **Weakness 1: The work is dependent on the periodic boundary conditions and simple dynamics, which create a situation where a predator can approach from two opposing directions. Further, predator-prey problems in 2D are quite the special case and may not be of huge generality. The line attractor property is not theoretically backed.**
>
> **Response 1**: Thanks for comments. Our study extends beyond periodic boundary conditions: First, we have also evaluated scenarios within **confined spaces**, as detailed in Appendices E and F.
>
> Second, the proposed EEC framework and ARM are not limited to pursuit-evasion scenarios but are also applicable to **cooperative tasks without adversaries**. A prime example is **multi-robot shape assembly**, where agents must populate a global target shape while maintaining uniform spacing and avoiding collisions. As shown in Figure 8, ARM successfully reveals that the agents have learned the underlying geometric structure. Initially, the unoccupied target center exhibits high Critic values, drawing agents inward. As the center fills, the value peak shifts to the boundary to promote the exploration of unoccupied regions. This dynamic mirrors the design intuition in Sun et al. (2023) and is supported by the quantitative analysis in Figure 8(c).
>
> These results suggest that **elucidating the mechanisms behind emergent collective behavior can provide new insights for MARL and inform the principled design of robust swarm-control policies**. Moreover, the applicability of our method extends beyond simple convex shapes to complex non-convex and multi-component geometries (e.g., the letter "B"). In Appendix F, we demonstrate a scenario where agents successfully assemble the letter "B" with ARM accurately capturing the learned geometric structure. This further reinforces the generalizability of our approach.
>
> Regarding the theoretical proof of line attractor property, we acknowledge that due to the data-driven and highly non-linear nature of neural networks, providing a rigorous theoretical proof is challenging. However, based on the fundamental definition of a stable attractor, verification relies on demonstrating that agents deviating from the attractor can successfully recover. Consequently, **validating the line attractor property is equivalent to verifying the agent's ability to recover from significant deviations**. To this end, we tested scenarios across nearly **20 different random seeds**. This statistical verification, based on a large sample size, serves as a "Monte Carlo-style" assessment of system stability, covering a broader range of the state space than analytical derivations.
>
> The boundary of the predators’ Voronoi diagram forms a line attractor is verified by **two different approaches**, shown in Figure 5 (b), (c), respectively. **The first** is manipulating predator positions to alter the Voronoi boundary geometry. **The second** is evaluating prey convergence under impulse perturbations against three baseline collective models.
> We have conducted perturbation experiments and compared our RL model against three baseline collective models  (Vicsek et al., 1995; Reynolds, 1987; Couzin et al., 2002). The results are **updated in Figure 5(c)** of the revised manuscript. We augmented the classic baselines with a predator-repulsion force to enable avoidance behavior.  To ensure a fair comparison, we optimized the control parameters for all baselines using a search algorithm to maximize their survival performance in this specific environment. An impulse perturbation is introduced to push prey agents away from the Voronoi boundar. As shown in Figure 5(c), the RL model demonstrates superior stability compared to the baselines:
> 1. **Steady-state Distance**:  The RL model achieves a steady-state distance of $0.05$, outperforming the baselines ($\approx 0.10$).
> 2. **Deviation distance**：the deviation distance of the RL model remains below $0.15$ and rapidly reconverges, whereas baseline deviations all exceed $0.15$.
>
> **Summary: These quantitative metrics empirically confirm the attractor property of the Voronoi boundary.**

---

> ### Author Response · Authors · 2025-12-03
> **Response to Reviewer HkEp (Weakness 2 &3)**
>
> > **Weakness 2: The extension to the multi-robot shape assembly task in Section 5 feels disconnected from the main analysis. The section is very brief, with most details relegated to the appendix. It does not clarify whether an analogous geometric structure (like the Voronoi diagram) emerges or how the EEC framework provides unique insights in that context.**
>
> **Response 2**: Thank you for your comments. In the revised manuscript, we have substantially expanded Section 6 to include detailed results regarding the multi-robot shape assembly task.   The proposed EEC framework **remains effective in revealing the geometric structures learned by robots** in this new task.
> We observe that when the target shape is not yet occupied, its center exhibits high critic values, drawing agents toward the center. Once the center becomes occupied, the value peak shifts to the boundary, promoting exploration of unoccupied areas.
>
> This finding is **supported by  the quantitative analysis** in Figure 8(c). While the internal mean value drops over time due to occupancy, the "**In-Shape Top** 10%" metric remains consistently higher than the "Out-Shape" values. This confirms that the Critic network successfully identifies the target interior as the region of highest global value. The stabilization of metrics around $t=50$ indicates convergence to a stable geometric attractor, validating the effectiveness of ARM in analyzing complex shape assembly.
>
> > **Weakness 3: The ARM is constructed using a "virtual probe" agent with zero velocity. While this is a clever way to isolate the policy's spatial response, a trained policy is conditioned on the full state, including velocity.**
>
> **Response 3**:
> Thank you for your comments. In our original submission, we evaluated the impact of **varying probe velocities** on the virtual probe, as shown in Appendix Figures 9–10. These initial tests indicated that velocity variations have minimal effect on the resulting ARM. In the revised manuscript, we have extended this sensitivity analysis to include **additional parameters, including heading angle, sampling resolution, agent counts, and noise interference** (Appendix Figures 8–17). The results demonstrate that ARM consistently reveals the underlying Voronoi diagram across these varying conditions, **confirming the method's robustness to parameter changes**.

---

> ### Author Response · Authors · 2025-12-03
> **Response to Reviewer HkEp (Question 1 &2)**
>
> > **Question 1: Could you elaborate on the role of the periodic boundary conditions in the emergence of the Voronoi diagram as an attractor? Have you run experiments in a large, un-walled environment or on other topologies? If so, does a similar geometric structure emerge, or do the agents adopt a different strategy?**
>
> **Response 4**:
> Thank you for your questions.
> First, we address the relationship between Voronoi diagrams and periodic boundaries:
>
> **Periodic boundary conditions do not inherently generate Voronoi structures**. For instance, in the flocking scenario presented in Appendix D, when no predator is present, prey agents do not aggregate near the Voronoi boundary, nor does their relative distance decrease. Without our analytic framework, the mechanism behind this lack of distance reduction might remain ambiguous. However, ARM clearly elucidates the cause: the relative orientation control, $a_{\mathrm{R}}$, is mostly distributed within the range of $[-0.2, 0.2]$, causing prey at different locations to adopt similar orientations. Simultaneously, the forward acceleration, $a_{\mathrm{F}}$, remains near its maximum value throughout the task space. Consequently, prey maintain similar velocities in both the $x$ and $y$ directions, resulting in no relative velocity differences and, therefore, no reduction in relative distance.
>
> Second, our EEC method is not limited to identifying Voronoi boundaries in competitive pursuit-evasion tasks, it is **applicable to cooperative tasks, such as shape assembly**. Unlike pursuit-evasion under periodic boundaries, shape-assembly tasks **require agents to fill a global target shape** while maintaining uniform spacing and avoiding collisions. We observe that when the target shape is initially unoccupied, its center exhibits high Critic values, drawing agents inward. Once the center becomes occupied, the value peak shifts to the boundary, promoting the exploration of unoccupied areas. This dynamic mirrors the design intuition in Sun et al. (2023) and supports the validity of our explanatory method.
>
> In conclusion, the proposed EEC framework and ARM are not confined to specific pursuit-evasion scenarios. They are **capable of analyzing diverse MARL settings ranging from competitive to cooperative tasks**, to uncover the distinct hidden geometric structures governing agent behavior.
>
>
> > **Question 2: Did you experiment with using a probe agent with non-zero (e.g., average) velocity? How sensitive is the structure of the ARM to the probe agent's own state, and would this change the interpretation of the results?**
>
> **Response 5**: Thank you for your question. We evaluated the performance of ARM under various parameter settings, including **different velocities** for the virtual probe agent. As shown in Appendix Figures 8–17, ARM consistently reveals the Voronoi diagram learned by the agents across these conditions. This confirms the robustness of ARM to parameter variations; for further details, please refer to **Response 3**.

---

> ### Author Response · Authors · 2025-12-03
> **Response to Reviewer HkEp (Question 3)**
>
> > **Question 3: The paper shows agents converge to this solution, but could you speculate more on why the MARL algorithm finds this specific Voronoi-based strategy? Is it demonstrably the most optimal path, or is it simply always the case for the simple problem dynamics with continuous policies?**
>
> **Response 6**: Thanks for your question. Actually, this RL model is an optimal escape strategy, which can be demonstrated by both experimental and theoretical approaches:
>
> 1. **Experimental Validation**:  We have conducted perturbation experiments and compared our RL model against **three baseline collective models**  (Vicsek et al., 1995; Reynolds, 1987; Couzin et al., 2002). The results are updated in Figure 5(c) of the revised manuscript. we augmented the classic baselines with a predator-repulsion force to enable avoidance behavior.  To ensure a fair comparison, we optimized the control parameters for all baselines using a search algorithm to maximize their survival performance in this specific environment. An impulse perturbation is introduced to push prey agents away from the Voronoi boundary. We repeated this experiment across **20 random seeds** to ensure statistical robustness. As shown in Figure 5(c), **the RL model demonstrates superior stability compared to the baselines**:
>     * **Steady-state Distance**:  The RL model achieves a steady-state distance of $0.05$, outperforming the baselines ($\approx 0.10$).
>     * **Deviation distance**：the deviation distance of the RL model remains below $0.15$ and rapidly reconverges, whereas baseline deviations all exceed $0.15$.
>
> Furthermore, we also evaluated the **survival rate** by comparing the average number of collisions over 20 random seeds:
> | Collective models | RL | Vicsek | Reynolds | Couzin |
> | :---: | :---: | :---: | :---: | :---: |
> | **Number of collisions** | 27 | 52 | 98 | 83 |
>
> The data indicates that the RL approach reduces the frequency of predator captures, thereby enhancing the prey's survival rate.
>
> 2. **Theoretical Optimality** of Voronoi Boundaries: The Voronoi boundary represents the theoretically optimal solution for risk minimization:
>     - **Biology (Selfish Herd)**: Extensive biological research models individual predation risk as a function of the Voronoi cell ("domain of danger") [1]. Moving toward the boundary of these domains is an evolutionary optimal strategy to shift risk to neighbors or maximize safety margins.
>     - **Robotics & Control (Optimal Path)**: In multi-obstacle or multi-threat environments, the "safest" path is to maximize the distance to the nearest obstacle/threat, which corresponds to the Voronoi boundary  [2,3]. Therefore, by converging to the Voronoi boundary, the RL agents are essentially approximating the optimal "Maximin" strategy (maximizing the minimum distance to any predator).
>
> However, while Voronoi-based control algorithms exist,  they are typically **hand-crafted with human priors**, making agents to calculate Voronoi cells [4], assuming the agents know the geometry beforehand. We do not know if biological evolution actually "computes" this or approximates it via other means. Furthermore, it assumes agents possess global geometric knowledge, and the real-time calculation of Voronoi boundaries is **computationally expensive, limiting scalability for large clusters**.
>
> **Our Contribution**: This paper **establishes a link between Voronoi-based evasion and evolutionary survival pressure**: Under a simple survival-pressure reward (without any geometric hints), Deep RL spontaneously discovers this optimal geometric structure. **This suggests that the Voronoi strategy is a natural attractor in the policy space for survival tasks**: Nature evolves this strategy because it is the most efficient way to survive, not because it was programmed to do so. These findings imply that complex collective behaviors in nature may be evolutionary responses to survival pressure and offer new inspiration for designing robust robot controllers.
>
> **References**
>
> [1] Hamilton, William D. "Geometry for the selfish herd." Journal of theoretical Biology 31.2 (1971): 295-311.
>
> [2] Choset, H., & Burdick, J. (2000). Sensor-based exploration: The hierarchical generalized voronoi graph. The International Journal of Robotics Research, 19(2), 96-125.
>
> [3] Bakolas, E., & Tsiotras, P. (2010). Optimal pursuit of moving targets using dynamic Voronoi diagrams. In 49th IEEE conference on decision and control (CDC) (pp. 7431-7436). IEEE.
>
> [4] Bakolas, E., & Tsiotras, P. (2010). The Zermelo–Voronoi diagram: A dynamic partition problem. Automatica, 46(12), 2059-2067.

---

### Official Review · Reviewer_PHNa · 2025-10-31

**Soundness:** 3
**Presentation:** 3
**Contribution:** 2
**Rating:** 4
**Confidence:** 3

**Summary:**

This paper investigates why complex swarm-like behaviors emerge in multi-agent reinforcement learning (MARL) systems trained only with simple survival-pressure rewards, without any explicit incentives for grouping. The authors propose a two-stage explanatory framework, EEC (Ego-observation → Ego-behavior → Collective behavior), and introduce a visualization tool called the Agent Response Map (ARM) to reveal how individual policies respond spatially across the environment. Through SHAP-based feature attribution and ablation analysis, they show that prey agents’ actions are dominated by the relative position of the nearest predator rather than by neighbors’ orientations. ARM further demonstrates that agents implicitly learn to align along the Voronoi boundaries of predators, forming a line attractor that explains aggregation and coordination. The framework is also applied to a robotic shape-assembly task to illustrate generality.

**Strengths:**

1. Clear and timely problem framing. The paper targets an interesting and underexplored question — explaining emergent collective behavior in MARL.

2. Interpretability framework. The combination of SHAP analysis and the proposed ARM visualization provides new insight into how local observations translate into swarm-level structure.

3. Mechanistic insight. Discovering that agents implicitly learn the predators’ Voronoi boundaries as safety zones is a compelling and intuitive explanation for emergent coordination.


4. Readable and well-structured. The paper is easy to follow and the figures effectively communicate the key findings.

**Weaknesses:**

1. Limited generality. The entire analysis is based on a predator–prey pursuit–evasion setting with homogeneous agents. It remains unclear whether the same mechanism explains other emergent behaviors in heterogeneous or task-driven MARL.

2. Primarily interpretive, not algorithmic. The work does not propose new learning methods or measurable performance improvements; its contribution is mainly analytical.

3. Dependence on specific geometric assumptions. The observed Voronoi-boundary alignment may arise from the planar and isotropic design of the environment, and might not hold in 3-D or irregular terrains.

4. Qualitative validation of ARM. Although ARM is visually compelling, its evaluation is descriptive rather than quantitative; a more formal measure of explanatory power would strengthen the claim.

5. Computational scalability. The framework requires repeated policy queries across the full spatial grid, which may become infeasible for high-dimensional environments.

**Questions:**

1. How sensitive are the ARM findings to the number of predators or to sensor noise in observation features?

2. Could the same EEC + ARM framework explain collective behaviors in cooperative tasks without adversaries (e.g., flock navigation or foraging)?

3. Is the “Voronoi line attractor” still observable when agents have non-Euclidean sensing or act in 3-D space?

4. Can ARM be extended to quantify causal influence rather than correlation？

---

> ### Author Response · Authors · 2025-12-03
> **Response to Reviewer PHNa (Weakness 1 &2)**
>
> > **Weakness 1: Limited generality. The entire analysis is based on a predator–prey pursuit–evasion setting with homogeneous agents. It remains unclear whether the same mechanism explains other emergent behaviors in heterogeneous or task-driven MARL.**
>
> **Response 1**: Thank you for your comments. The proposed explanatory framework and ARM can be generalized to various of MARL tasks besides pursuit-evasion. In our paper, we applied this framework to a **task-driven MARL** environment: multi-robot shape assembly. This is a purely cooperative task requiring agents to **fill a global target shape constraint while maintaining uniform spacing and avoiding collisions**.
>
> As shown in Figure 8, ARM successfully reveals that the agents **learn the underlying geometric structure of the target**. Initially, when the target shape is unoccupied, its center exhibits high critic values, which draws agents inward. Once the center becomes occupied, the value peak shifts to the boundary, promoting the exploration of unoccupied areas. This dynamic mirrors the design intuition in Sun et al. (2023) and is further supported by the quantitative analysis in Figure 8(c). These results validate the effectiveness of ARM in identifying meaningful geometric invariants in task-driven cooperative MARL.
>
> Furthermore, the applicability of our method extends beyond simple convex shapes to **complex non-convex and multi-component geometries** (e.g., English letters). In Appendix F, we demonstrate a scenario where agents assemble the letter "B". The results confirm that agents successfully achieve the target formation, and ARM accurately captures the learned geometric structure in this complex setting, further reinforcing the generalizability of our approach.
>
>
> > **Weakness 2: Primarily interpretive, not algorithmic. The work does not propose new learning methods or measurable performance improvements; its contribution is mainly analytical.**
>
> **Response 2**: Thanks for your comments. Although the performance-driven methods (Black-box) push state-of-the-art results, they often lack transparency, making it difficult to diagnose failures or guarantee safety in real-world deployments.
> Our work addresses this gap by **proposing the EEC framework to transform these opaque policies into a "white box".** We introduce the Agent Response Map (ARM), a novel diagnostic method that reveals the hidden link between Ego-behavior and Collective-behavior. Specifically, it uncovers that agents implicitly learn and exploit geometric structures to achieve coordination, which is a mechanism previously unknown.
>
>
> **Beyond Qualitative Description to Quantitative Analysis**: In the revision, we have more quantitative analysis in Figure 5, to move beyond visual intuition:
> - **Dynamics Quantification**: We quantified the "**Steady-state Distance**" and "**Deviation Distance**" to prove the Voronoi boundary acts as a stable line attractor, and we benchmarked this against classic baseline models (Vicsek, Reynolds, Couzin) (Figure 5 (c)).
> - **Discontinuity Metrics**: We introduced a **spatial gradient metric** for ARM to quantitatively measure the discontinuity boundaries, confirming that the learned geometric features are statistically significant and robust across seeds (Figure 5 (a)).
> - **Predictive Accuracy**: We verified that the ARM-derived gradient maps can accurately predict agent aggregation zones in actual rollouts (**Concentration Ratio** = 1.52x).
>
> **Alignment with Community Standards**: The mechanistic interpretability is a well-established and highly valued track in top-tier AI conferences. Recent works [1, 2, 3] published on previous ICLR all focus on explaining why agents behave the way they do rather than performance improvements.   **Our work aligns with this important direction by providing the first mechanistic explanation for emergent collective behavior in MARL**.
>
> **References**
>
> [1] Thomas Bush, Stephen Chung, Usman Anwar, Adrià Garriga-Alonso, and David Krueger. Interpreting emergent planning in model-free reinforcement learning. In The Thirteenth International Conference on Learning Representations, 2025.
>
> [2] Shripad Vilasrao Deshmukh, Arpan Dasgupta, Balaji Krishnamurthy, Nan Jiang, Chirag Agarwal, Georgios Theocharous, Jayakumar Subramanian. Explaining RL Decisions with Trajectories. In The Eleventh International Conference on Learning Representations, 2023.
>
> [3] Hyunju Kang, Geonhee Han, Hogun Park. UNR-Explainer: Counterfactual Explanations for Unsupervised Node Representation Learning Models.  In The Twelfth International Conference on Learning Representations, 2024.

---

> ### Author Response · Authors · 2025-12-03
> **Response to Reviewer PHNa (Weakness 3)**
>
> > **Weakness 3: Dependence on specific geometric assumptions. The observed Voronoi-boundary alignment may arise from the planar and isotropic design of the environment, and might not hold in 3-D or irregular terrains.**
>
> **Response 3**: Thanks for your comments. Actually, our results are **robustness to environmental asymmetry**: To refute the reliance on perfect environmental isotropy, we introduced **perceptual perturbation** by injecting noise into the agents' observations. The noise breaks the perfect symmetry of the environment, simulating the uncertainty and errors characteristic of "irregular terrains" or sensor degradation.
> - **Results**: As shown in Appendix Figure 11-12, even under significant perceptual distortion, agents still successfully converge to the Voronoi boundary. This empirical evidence proves that the Voronoi attractor is **robust to environmental imperfections** and is not merely an artifact of an idealized setup.
>
> The Voronoi boundary mechanism can be extended to 3D Euclidean space.
> - **Mathematical Universality**: Theoretically, the concept of Voronoi partitioning generalizes naturally to higher dimensions and non-Euclidean metrics. In 3D space, the "line attractor" extends to a "surface attractor" (Voronoi faces. This is a standard construction extensively reviewed in Okabe et al. (2000)).
> - **Pursuit-evasion**: Ref.[1] explicitly construct a Voronoi diagram in 3D flow fields to partition the space into dominance regions of different pursuers, with the boundaries defined by equal-time-to-capture manifolds.
> - **Real animal collectives**: Recent 3D models of bait-ball formation rely on Voronoi neighborhoods to reproduce realistic three-dimensional group structures to simulating the fish schooling [3].
>
> Our framework is **scalable and feasible** for high-dimensional environments (e.g., 3D) for two reasons:
> - **Offline Analysis**: ARM analysis is performed post-hoc, meaning it does not add overhead to the online RL training process.
> - **Parallel Efficiency**: The generation of ARM is highly parallelizable. We can batch the entire spatial grid into the GPU for single-pass inference.
>
> We conducted a runtime analysis (**averaged over 20 seeds**) comparing 2D vs. 3D ARM generation at different sampling resolutions. As shown in Table below, reducing resolution significantly accelerates computation while retaining the ability to identify geometric structures.
> | Sampling Resolution | 100 | 80 | 50 |
> | :---: | :---: | :---: | :---: |
> | **2D Space** | 1.6 | 1.08 | 0.52 |
> | **3D Space** | 3.5 | 2.1 | 1.2 |
>
> **Generalization to Task-Driven MARL**: Our framework is not limited to the pursuit-evasion domain. It serves as a **general diagnostic tool for broader MARL tasks**. For example, we applied the explanation  framework to a **robotic shape assembly task** (see Section 6). In this completely different setting (cooperative, global constraint), ARM successfully identified the implicit geometric potential fields driving the agents to form complex shapes. This demonstrates the method's versatility and potential for practical multi-robot coordination.
>
> **References**
>
> [1] Sun, W., Tsiotras, P., & Yezzi, A. J. (2019). Multiplayer pursuit-evasion games in three-dimensional flow fields. Dynamic Games and Applications, 9(4), 1188-1207.
>
> [2] Liu, D., Liang, Y., Deng, J., & Zhang, W. (2023). Modeling three-dimensional bait ball collective motion. Physical Review E, 107(1), 014606.

---

> ### Author Response · Authors · 2025-12-03
> **Response to Reviewer PHNa (Weakness 4 &5)**
>
> > **Weakness 4: Qualitative validation of ARM. Although ARM is visually compelling, its evaluation is descriptive rather than quantitative; a more formal measure of explanatory power would strengthen the claim.**
>
> **Response 4**: Thank you for your comments. In the original submission, we employed a qualitative evaluation to validate the ARM interpretation results, specifically using the **relative distance between prey and the Voronoi boundary** to verify its nature as a line attractor. In the revised manuscript, we have expanded this evaluation framework to support our claims through three distinct approaches:
> 1. **Attractor Verification**: We verify that the Voronoi boundary functions as a line attractor by **two** different approaches: The first is manipulating predator positions to alter the Voronoi boundary geometry. The second is evaluating prey convergence under impulse perturbations against three baseline collective models.
> 2. **Predictive Validation**: We utilize ARM-identified features to predict agent aggregation in actual rollouts, thereby quantifying **causal influence** rather than mere correlation.
> 3. **Sensitivity Analysis**: We quantify performance stability under variations in agent motion parameters.
>
> To rigorously evaluate these approaches, we introduced the following three quantitative metrics:
> - **Dynamics Quantification**: We calculated the "**Steady-state Distance**" and "**Deviation Distance**" to demonstrate that the Voronoi boundary acts as a stable line attractor. We benchmarked these results against classic collective models (Vicsek, Reynolds, Couzin), as shown in Figure 5(c).
> - **Discontinuity Metrics**: We introduced a **spatial gradient metric** for ARM to quantitatively measure discontinuity boundaries. This confirms that the learned geometric features are statistically significant and robust across random seeds (Figure 5(a)).
> - **Predictive Accuracy**: We verified that the ARM-derived gradient maps can accurately predict agent aggregation zones in actual rollouts, achieving a **concentration ratio** = 1.52x.
> These details, presented in Figure 5, confirm the boundary's attractor property and suggest that the RL-learned policy approximates an optimal strategy.
>
>
> Furthermore, for the task-driven multi-robot shape assembly scenario, the quantitative analysis in Figure 8(c) further supports our findings. While the internal mean value drops over time due to occupancy, the "**In-Shape Top** 10%" metric remains consistently higher than the "Out-Shape" values. This confirms that the Critic network successfully identifies the target interior as the region of highest global value. The stabilization of metrics around $t=50$ indicates convergence to a stable geometric attractor, validating the effectiveness of ARM in analyzing complex shape assembly.
>
> In summary, **this comprehensive quantitative analysis validates that ARM remains effective in identifying meaningful geometric attractors across various MARL tasks**, thereby demonstrating the generalizability of our method.
>
>
> > **Weakness 5: Computational scalability. The framework requires repeated policy queries across the full spatial grid, which may become infeasible for high-dimensional environments.**
>
> **Response 5**: Thanks for your suspicion. Actually, our method can be extended to a higher dimension space  for two reasons:
> - **Offline Analysis**: The ARM analysis is performed post-hoc, meaning it does not add overhead to the online RL training process.
> - **Parallel Efficiency**: The generation of ARM is highly parallelizable. We can batch the entire spatial grid into the GPU for single-pass inference.
>
> We conducted a runtime analysis **(averaged over 20 seeds)** comparing 2D vs. 3D ARM generation at different sampling resolutions. As shown in Table R1 below, reducing resolution can accelerate computation while retaining the ability to identify geometric structures.
>
> | Resolution | 100 | 80 | 50 |
> | :---: | :---: | :---: | :---: |
> | **2D Space** | 1.6 | 1.08 | 0.52 |
> | **3D Space** | 3.5 | 2.1 | 1.2 |

---

> ### Author Response · Authors · 2025-12-03
> **Response to Reviewer PHNa (Question 1-4)**
>
> > **Question 1: How sensitive are the ARM findings to the number of predators or to sensor noise in observation features?**
>
> **Response 6**:  Thank you for your question. In the revised manuscript, we conducted additional evaluations of ARM results under varying **predator/prey population sizes** and **different measurement noise levels**, as shown in Appendix Figures 11–14.  The results indicate that while altering the number of agents changes the specific geometry of the Voronoi diagram, ARM remains capable of accurately characterizing these structures.  Similarly, while different noise amplitudes affect the magnitude of policy outputs (shifting the color scale) and influence specific pursuit-evasion dynamics, ARM retains its ability to reveal the fundamental Voronoi structure.
>
> Hence,  regardless of parameter variations, the **ARM consistently captures stable spatial patterns, demonstrating its robustness for interpreting swarm behavior**.
>
> > **Question 2: Could the same EEC + ARM framework explain collective behaviors in cooperative tasks without adversaries (e.g., flock navigation or foraging)?**
>
> **Response 7**:
> Thank you for your question. In fact, our paper already addresses two distinct scenarios without adversaries:
>
> 1. **Multi-Robot Shape Assembly**: This can be viewed as a variant of collective navigation, where the target shape represents the desired goal. Agents must populate this shape while avoiding collisions and preserving a uniform formation. Our proposed explanation method reveals that when the target shape is initially unoccupied, its center exhibits high Critic values, drawing agents inward. Once the center becomes occupied, the value peak shifts to the boundary, promoting the exploration of unoccupied areas. **This behavior mirrors the design intuition found in Sun et al. (2023) and supports the validity of our explanatory method**. Detailed information regarding this scenario is provided in Section 6.
> 2. **Flocking**: We also analyzed the formation of collective strategies in a flocking scenario. The primary difference from the predator-present scenario is that while flocking agents maintain consistent orientation, their relative distances do not continuously decrease. This phenomenon is effectively explained by the proposed ARM method, as detailed in Appendix D.
>
> > **Question 3: Is the “Voronoi line attractor” still observable when agents have non-Euclidean sensing or act in 3-D space?**
>
> **Response 8**: Thanks for your question. Our results are robustness to environmental asymmetry, and scalable and feasible for high-dimensional environments (e.g., 3D). We refer to **Response 3** for details.
>
>
> > **Question 4: Can ARM be extended to quantify causal influence rather than correlation?**
>
> **Response 9**: Thanks for your question. Actually ARM can be used to quantify causal influence. Establishing causality can be interpreted as using the Voronoi structure to **predict agent behavior and subsequently measuring the prediction accuracy.**
>
> First, leveraging the numerical discontinuities inherent in the Voronoi diagrams, we propose the **Spatial Gradient Magnitude (SGM)** of ARM. This metric quantifies discontinuities by calculating the gradient magnitude of ARM outputs with respect to spatial position. The resulting discontinuity map, shown in Figure 5(a), reveals that discontinuity peaks near the Voronoi boundaries, which is consistent with our ARM analysis.
>
> Based on this, we hypothesize that if the ARM-derived Voronoi diagram can predict prey agent behavior (i.e., agents tend to move toward directions with high ARM gradient values), then the ARM gradient values at the agents' actual locations should be significantly higher than those at random locations. This would confirm that agents aggregate toward the Voronoi boundaries.
>
> We introduce two metrics and test over 20 random seeds: the **Concentration Ratio** (the ratio of the mean gradient at agent locations to the global mean gradient) and the **Percentile Rank** (the ranking of agent-location gradients within the global distribution).
>
> 1. **Concentration ratio**: the gradient at agent locations is $1.52$x higher than the global baseline, which demonstrates that ARM accurately identifies the high-gradient regions that act as attractors. The agents are not moving randomly; they are actively congregating in the specific areas predicted by ARM.
>
> 2. **Percentile rank**: agents  consistently aggregate in the top $10.2$% of high-gradient regions, indicating high predictive precision. This indicates high predictive precision: **ARM successfully filters out near 90% "low-relevance" space, correctly pinpointing the narrow "corridors" where aggregation occurs**.
>
> These results indicate that the ARM-derived Voronoi boundaries serve as an effective predictor of agent behavior with high predictive precision.

---

### Official Review · Reviewer_aMWe · 2025-11-04

**Soundness:** 3
**Presentation:** 3
**Contribution:** 3
**Rating:** 6
**Confidence:** 3

**Summary:**

This paper investigates why complex collective behaviors can spontaneously emerge in multi-agent reinforcement learning (MARL) environments even when agents are only trained with simple survival/catching rewards. The key contributions are:
1. The paper identifies an intuitive yet unexpected mechanism: agents implicitly learn the geometric risk field of the environment, especially the Voronoi boundaries induced by predators, and use these boundaries as coordination targets. Consequently, agents spontaneously gather along these boundaries, which act as linear attractors. This finding is supported through visualization and empirical analysis.
2. The authors propose a two-stage explanatory pipeline EEC. Stage 1 identifies key sensory inputs via SHAP-based attribution and controllability tests. Stage 2 introduces the Agent Response Map (ARM), a new tool using a virtual invisible probe to scan space, visualize policy responses, and reveal spatial discontinuities and agents’attractor lines.
3. The method is further demonstrated in a shape-assembly task with robotic agents, showing that the framework generalizes beyond the predator–prey domain and has potential for practical multi-robot coordination.

**Strengths:**

Originality is strong. The idea that agents implicitly learn and exploit geometric invariants (Voronoi boundaries) is novel and not widely explored in MARL interpretability. The ARM tool is a new methodological contribution that bridges local policy interpretation with global spatial structure.

Quality is good, but could be stronger. The authors combine attribution analysis (SHAP), ablation tests, and spatial visualization (ARM) effectively. The experiments cover multiple scenarios—bounded vs. periodic maps, predator vs. no-predator, and transfer to robot shape assembly. However, the main claim (“Voronoi boundaries act as attractors”) is still mostly supported by visual trends, not rigorous quantitative evidence. The results are suggestive but need more formal validation (e.g., dynamical analysis, statistical robustness).

Clarity is above average. The main storyline is clear and supported by good visuals (ARM heatmaps, SHAP plots, time–distance graphs). The EEC pipeline is conceptually clean. That said, several implementation details (e.g., network structure, hyperparameters, training seeds, ARM sampling strategy) are deferred to the appendix or missing altogether, which makes reproducibility difficult. Moreover, some claims (like “implicit Voronoi learning”) would benefit from clearer quantitative definitions.

Significance is high. Understanding why emergent collective behavior arises under simple rewards is a long-standing question in MARL, robotics, and swarm intelligence. If validated, the proposed mechanism could reshape how we interpret learned coordination and provide a useful diagnostic tool (ARM) for multi-agent systems.

**Weaknesses:**

Lack of quantitative proof for the “line attractor” / Voronoi learning claim

Suggetions:

1. Attractor Quantification: Introduce perturbation experiments around Voronoi boundaries. Measure return-to-boundary time and convergence speed compared with baselines (random policies, Vicsek-style local rules). If Voronoi boundaries are true attractors, trajectories should converge faster and more consistently.

2. Local Linearization: Compute local Jacobians of the policy-induced velocity field along the boundary (e.g., via finite differences) to show contraction in the perpendicular direction.

3. Alternative Boundary Comparison: Compare Voronoi boundaries with alternative geometric features (e.g., iso-value contours of the critic or potential field) to demonstrate that Voronoi is indeed the most predictive structure.

Insufficient robustness and quantitative metrics for ARM

Suggestions:

1. Conduct parameter sensitivity analysis for ARM: vary the probe’s initial orientation, speed, neighbor count, and input ordering to verify that results are consistent.

2. Define a quantitative metric for ARM discontinuity (e.g., spatial gradient variance) and report it across seeds to establish reproducibility.

3. Perform predictive validation: use ARM maps to predict where agents will aggregate in actual rollouts and measure the prediction accuracy.

Missing Baseline Comparisons

Suggestions:

1. Compare against classical local-interaction models (Vicsek, Couzin, Reynolds) trained or tuned under the same reward conditions to test whether similar Voronoi clustering emerges.

2. Extend the explanation method comparison beyond SHAP: e.g., causal feature ablation (random permutation or noise injection) to confirm that input importance reflects causality, not just correlation.

Reproducibility Gaps

Suggestions:

1. Provide full training details (architecture, optimizer, learning rate, batch size, seeds, hardware, total steps).

2. Release code or demo scripts to ensure transparency and facilitate community adoption.

Limited discussion on generalization and real-world constraints

Suggestions:
1. Expand shape-assembly experiments to non-convex or multi-component target shapes and report whether ARM still identifies meaningful geometric attractors.

2. Discuss (or test) how occlusion, noise, or communication delay affects the learned geometry—important for practical multi-robot coordination.

**Questions:**

1. Can you provide statistics on how agents return to Voronoi boundaries after perturbation, compared with random baselines?

2. How sensitive are ARM maps to probe orientation, initial position, or sampling resolution?

3. How are Voronoi boundaries computed under periodic boundary conditions or multiple predators? Are they always continuous?

4. How many random seeds were used? What is the variance in emergent behavior across runs?

5. How do results change with different prey/predator ratios or varying perceptual neighbor counts?

6. How does your learned strategy differ quantitatively from rule-based models (in clustering degree, survival rate, or energy efficiency)?

---

> ### Author Response · Authors · 2025-12-03
> **Response to Reviewer aMWe (Weakness 1 &2)**
>
> > **Weakness 1: Attractor Quantification: Introduce perturbation experiments around Voronoi boundaries. Measure return-to-boundary time and convergence speed compared with baselines (random policies, Vicsek-style local rules). If Voronoi boundaries are true attractors, trajectories should converge faster and more consistently.**
>
>
> **Response 1**: Thanks for your advice. We have conducted perturbation experiments and compared our RL model against **three baseline collective models**  (Vicsek et al., 1995; Reynolds, 1987; Couzin et al., 2002). The results are updated in Figure 5(c) of the revised manuscript.
>
> We augmented the classic baselines with a predator-repulsion force to enable avoidance behavior.  To ensure a fair comparison, we optimized the control parameters for all baselines using a search algorithm to maximize their survival performance in this specific environment. An impulse perturbation is introduced to push prey agents away from the Voronoi boundary. We repeated this experiment across **20 random seeds** to ensure statistical robustness. As shown in Figure 5(c), the RL model demonstrates superior stability compared to the baselines:
> 1. **Steady-state Distance**:  The RL model achieves a steady-state distance of $0.05$, outperforming the baselines ($\approx 0.10$).
> 2. **Deviation distance**：the deviation distance of the RL model remains below $0.15$ and rapidly reconverges, whereas baseline deviations all exceed $0.15$.
>
> These quantitative metrics empirically confirm the attractor property of the Voronoi boundary.
>
>
> > **Weakness 2: Local Linearization: Compute local Jacobians of the policy-induced velocity field along the boundary (e.g., via finite differences) to show contraction in the perpendicular direction.**
>
> **Response 2**: Thank you for your advice. Actually, the policy network of our RL model is highly non-linear. Hence,  local linearization (via the Jacobian matrix) can only reflect properties within an infinitesimal neighborhood. However, to demonstrate that the Voronoi boundary serves as an effective line attractor, it is more critical to **verify whether the agent can recover when deviating far from the boundary (i.e., under large-scale conditions)**.
>
> On the other hand, the RL model is a data-driven approach; therefore, obtaining the Jacobian matrix for the agent at different positions is essentially performing numerical simulations at those locations, rather than deriving analytical solutions. This differs from traditional rule-based models, which can calculate the analytical solution of the Jacobian matrix directly from formulas without the need for specific simulations.
>
>
> Therefore, to prove that the Voronoi boundary is an effective attractor  is to **verify the agent's ability to recover from large deviations**. We validated the convergence of prey agents at the Voronoi boundary through the perturbation experiments described in **Response 1**. We tested scenarios across nearly **20 different random seeds**. This statistical verification based on a large sample size serves as a "Monte Carlo-style" assessment of system stability, covering a broader range of the state space than single Jacobian calculations.

---

> ### Author Response · Authors · 2025-12-03
> **Response to Reviewer aMWe (Weakness 3 - 5)**
>
> > **Weakness 3: Alternative Boundary Comparison: Compare Voronoi boundaries with alternative geometric features (e.g., iso-value contours of the critic or potential field) to demonstrate that Voronoi is indeed the most predictive structure.**
>
> **Response 3**: Thank you for the suggestion. This idea can be interpreted as a proposal to extend the ARM to quantify causal influence rather than correlation.  In other words, we aim to use the Voronoi structure to **predict agent behavior**.
>
>
> First, leveraging the numerical discontinuities inherent in the Voronoi diagrams, we propose the **Spatial Gradient Magnitude (SGM)** of ARM. This metric quantifies discontinuities by calculating the gradient magnitude of ARM outputs with respect to spatial position. The resulting discontinuity map, shown in Figure 5(a), reveals that discontinuity peaks near the Voronoi boundaries, which is consistent with our ARM analysis.
>
> Based on this, we hypothesize that if the ARM-derived Voronoi diagram can predict prey agent behavior (i.e., agents tend to move toward directions with high ARM gradient values), then the ARM gradient values at the agents' actual locations **should be higher than those at random locations**. This would confirm that agents aggregate toward the Voronoi boundaries.
>
> We introduce two metrics and test over **20 random seeds**: the Concentration Ratio (the ratio of the mean gradient at agent locations to the global mean gradient) and the Percentile Rank (the ranking of agent-location gradients within the global distribution).
> 1. **Concentration ratio**: the gradient at agent locations is **$1.52$x** higher than the global baseline, which demonstrates that ARM accurately identifies the high-gradient regions that act as attractors. The agents are not moving randomly; they are actively congregating in the specific areas predicted by ARM.
> 2. **Percentile rank**: agents  consistently aggregate in the top $10.2$% of high-gradient regions, indicating high predictive precision. This indicates high predictive precision: **ARM successfully filters out near 90% "low-relevance" space, correctly pinpointing the narrow "corridors" where aggregation occurs.**
>
> These results indicate that the ARM-derived Voronoi boundaries serve as an effective predictor of agent behavior with high predictive precision.
>
> > **Weakness 4: Conduct parameter sensitivity analysis for ARM: vary the probe’s initial orientation, speed, neighbor count, and input ordering to verify that results are consistent.**
>
> **Response 4**: Thank you for your suggestion. In our original submission, we evaluated the sensitivity of ARM to various probe initialization factors, including **orientation, speed, and different random seeds** (see Appendix Figures 8–10). The impact of perceptual neighbor counts was also analyzed in Appendix Figure 3.
> In the revised manuscript, we have extended this evaluation to include **varying population sizes, sampling resolutions, and measurement noise levels**. These additional results are presented in Appendix Figures 11–16. Across all tested parameter settings, ARM consistently reveals the underlying Voronoi structure, confirming the method's robustness to parameter variations.
>
> > **Weakness 5: Define a quantitative metric for ARM discontinuity (e.g., spatial gradient variance) and report it across seeds to establish reproducibility.**
>
> **Response 5**: Thanks for your advice: We introduce the ARM **Spatial Gradient Magnitude (SGM)** to quantify discontinuities within the ARM results. Specifically, treating the ARM output as a scalar field, we calculate the gradient magnitude of the ARM output with respect to spatial position. The Discontinuity Metric is defined as $M = \lVert\nabla a_{\mathrm{R}}\rVert = \sqrt{(\frac{\partial a_{\mathrm{R}}}{\partial x})^2 + (\frac{\partial a_{\mathrm{R}}}{\partial y})^2}$. This effectively maps multi-directional neighborhood variations into a unified scalar measure of discontinuity, where **higher values indicate more drastic policy shifts**. This approach draws inspiration from edge detection and saliency map techniques in computer vision.
> As shown in Figure 5(a), the discontinuity peaks near the Voronoi boundaries where **the magnitude reaches approximately** $40$, whereas it remains below $5$ across the vast majority of the environment. Furthermore, we validated the consistency of the Spatial Gradient Magnitude across 20 different random seeds, with representative results provided in Appendix Figure 21. These findings demonstrate the robustness of our proposed method.

---

> ### Author Response · Authors · 2025-12-03
> **Response to Reviewer aMWe (Weakness 6 &7)**
>
> > **Weakness 6：Perform predictive validation: use ARM maps to predict where agents will aggregate in actual rollouts and measure the prediction accuracy.**
>
> **Response 6**: Thank you for your comments. Leveraging the Spatial Gradient Magnitude (SGM) established in Response 4, we can predict prey agent behavior. Specifically, we define two metrics averaged over 20 random seeds: the **Concentration Ratio** (the ratio of the mean gradient at agent locations to the global mean gradient) and the **Percentile Rank** (the ranking of agent-location gradients within the global distribution).
> As detailed in Response 3, our results show that **the gradient at agent locations is** $1.52\times$ **higher than the global baseline, while agents consistently aggregate within the top** $10.2$% **of high-gradient regions**. These findings indicate that the ARM-derived Voronoi boundaries serve as an effective predictor of agent behavior with high predictive precision.
>
>
>
> > **Weakness 7: Compare against classical local-interaction models (Vicsek, Couzin, Reynolds) trained or tuned under the same reward conditions to test whether similar Voronoi clustering emerges.**
>
> **Response 7**:
> Thank you for your advice. We compared our trained RL model against three distinct collective motion models: Vicsek, Couzin, and Reynolds under identical scenarios (Figure 5(c)). The results demonstrate that the RL model achieves the **fastest convergence and superior overall performance**. Furthermore, we evaluated **the survival rate** by comparing the average number of collisions over 20 random seeds:
> | Collective models | RL | Vicsek | Reynolds | Couzin |
> | :--- | :---: | :---: | :---: | :---: |
> | Number of collisions | 27 | 52 | 98 | 83 |
>
> The results demonstrate that the RL model exhibits the minimum number of collisions, corresponding to the highest survival rate among the evaluated methods.
>
> The RL model demonstrates optimal performance for the following two reasons:
> 1. **Limitations of Rule-Based Models**: Traditional models like Vicsek, Couzin, and Reynolds were proposed for specific contexts (e.g., phase transitions, information transfer, or animation). They rely on explicit, hand-crafted heuristics, which may not be theoretically optimal  in pursuit-evasion scenarios, as evidenced by the lower performance in Figure 5(c).
>
> 2. **Theoretical Optimality of Voronoi Boundaries**: Conversely, the Voronoi boundary represents the theoretically optimal solution for risk minimization:
>     - **Biology (Selfish Herd)**: Extensive biological research models individual predation risk as a function of the Voronoi cell ("domain of danger") [1]. Moving toward the boundary of these domains is an evolutionary optimal strategy to shift risk to neighbors or maximize safety margins.
>     - **Robotics & Control (Optimal Path)**: In multi-obstacle or multi-threat environments, the "safest" path is to maximize the distance to the nearest obstacle/threat, which corresponds to the Voronoi boundary  [2,3]. Therefore, by converging to the Voronoi boundary, the RL agents are essentially approximating the optimal "Maximin" strategy (maximizing the minimum distance to any predator).
>
> However, while Voronoi-based control algorithms exist,  they are typically hand-crafted with human priors, making agents to calculate Voronoi cells [4], assuming the agents know the geometry beforehand. We do not know if biological evolution actually "computes" this or approximates it via other means. Furthermore, it assumes agents possess global geometric knowledge, and the real-time calculation of Voronoi boundaries is computationally expensive, limiting scalability for large clusters.
>
> **Our Contribution**: This paper **establishes a link between Voronoi-based evasion and evolutionary survival pressure**: Under a simple survival-pressure reward (without any geometric hints), Deep RL spontaneously discovers this optimal geometric structure. **This suggests that the Voronoi strategy is a natural attractor in the policy space for survival tasks**: Nature evolves this strategy because it is the most efficient way to survive, not because it was programmed to do so. These findings imply that complex collective behaviors in nature may be evolutionary responses to survival pressure and offer new inspiration for designing robust robot controllers.
>
> **References**
>
> [1] Hamilton, William D. "Geometry for the selfish herd." Journal of theoretical Biology 31.2 (1971): 295-311.
>
> [2] Choset, H., & Burdick, J. (2000). Sensor-based exploration: The hierarchical generalized voronoi graph. The International Journal of Robotics Research, 19(2), 96-125.
>
> [3] Bakolas, E., & Tsiotras, P. (2010). Optimal pursuit of moving targets using dynamic Voronoi diagrams. In 49th IEEE conference on decision and control (CDC) (pp. 7431-7436). IEEE.
>
> [4] Bakolas, E., & Tsiotras, P. (2010). The Zermelo–Voronoi diagram: A dynamic partition problem. Automatica, 46(12), 2059-2067.

---

> ### Author Response · Authors · 2025-12-03
> **Response to Reviewer aMWe (Weakness 8-10)**
>
> > **Weakness 8: Extend the explanation method comparison beyond SHAP: e.g., causal feature ablation (random permutation or noise injection) to confirm that input importance reflects causality, not just correlation.**
>
> **Response 8**: Thank you for your advice. The most fundamental method to verify the causal influence of an input feature on network output is through intervention experiments. Specifically, this involves masking or fixing a specific feature within the observation to determine whether the RL agent can still effectively learn collective behaviors. In our paper, we have already conducted causal ablation experiments to validate the SHAP attribution results, as shown in Figure 3(c) and (d). These experiments **confirm that the identified input importance reflects true causality**.
> 1. **High-Importance Feature (Relative Position)**:
>       - **Causal Ablation**: When we removed relative position information (masked as zeros), the collective behavior completely collapsed (see Appendix B.1). This confirms the decisive causal role of relative position.
> 2. **Low-Importance Features (Relative Orientation & Acceleration)**:
>       - **Causal Ablation**: Conversely, when we removed orientation features (see Figure 3(c) ) or fixed the acceleration output to a constant (see Figure 3(d)), the swarming behavior remained robust and intact. This confirms their lack of causal necessity.
> To ensure statistical robustness and rule out chance, we repeated each ablation experiment across **20 different random seeds**. We observed consistent outcomes in all trials: the swarm consistently failed without position information and succeeded without orientation or acceleration control. **These results demonstrate that our SHAP analysis accurately reflects the true causal logic of the learned policy rather than mere correlation**.
>
> Furthermore, we compared our results against different feature-attribution baselines, including **Saliency Maps** and **Integrated Gradients**. The results were consistent with the SHAP analysis, as detailed in Appendix B.2.
>
> Finally, to better clarify the purpose of these ablation experiments for the reader, we have revised the relevant description in the main text (see Lines 311–321 in the revised paper).
>
>
> > **Weakness 9: Provide full training details (architecture, optimizer, learning rate, batch size, seeds, hardware, total steps).**
>
> **Response 9**: Thank you for your advice. In the original submission, technical details such as the network architecture, learning rate, batch size, and total steps were listed in the Appendix. In the revised manuscript, we have moved these **detailed RL training and network architecture specifications to the main text** (Page 4, Line 210). Additionally, we have included further specifications regarding the optimizer, hardware infrastructure, and training duration. These revisions ensure that the experimental results can be fully reproduced from scratch.
>
> > **Weakness 10: Release code or demo scripts to ensure transparency and facilitate community adoption.**
>
> **Response 10**: Thank you for your advice. We have provided the source code for RL training in the original submission. Furthermore, we are committed to making the codebase fully open-source to ensure reproducibility and facilitate future research.

---

> ### Author Response · Authors · 2025-12-03
> **Response to Reviewer aMWe (Weakness 11 &12)**
>
> > **Weakness 11: Expand shape-assembly experiments to non-convex or multi-component target shapes and report whether ARM still identifies meaningful geometric attractors.**
>
> **Response 11**: Thank you for your suggestion. In the revised paper, we have conducted additional tests on a new target shape, the letter "B", which represents a complex geometry featuring both non-convexity and multi-component structures.
>
> The results, supported by quantitative validation, are presented in Appendix Figures 33 and 34. Agents successfully transitioned from a random initial distribution to the target formation, demonstrating the method’s applicability to complex scenarios.
> In Figure 34(a), due to the random initial distribution, most agents are located outside the target shape. Consequently, the Critic value is highest around the shape's perimeter and gradually decreases as the distance increases; this reveals that **the agents have learned the geometric model of the target shape**. Conversely, although some agents occupy part of the target shape (bottom right) at initialization, the Critic value there is low. This indicates overcrowding, implicitly signaling that these **agents should disperse to maximize the global value**. Once the agents have assembled the shape (Figure 34(b)), the Critic value in the center becomes low because the space is fully occupied. However, the value at the edges remains relatively high, **encouraging agents to explore and fill these unoccupied boundary regions**.
>
> This observation mirrors the intuition in Sun et al. (2023) and supports the validity of our explanatory method.
> Quantitative analysis in panel (c) further supports this. While the internal mean value drops over time due to occupancy, the **"In-Shape Top 10%"** metric remains consistently higher than the "Out-Shape" values. This confirms that the Critic network successfully identifies the target interior as the region of highest global value. The stabilization of metrics around $t=50$ indicates **convergence to a stable geometric attractor**, validating the effectiveness of ARM in analyzing complex shape assembly.
>
>
> The above analysis validates that ARM remains effective in identifying meaningful geometric attractors in task-driven cooperative MARL, thereby demonstrating the generalizability of our method.
>
> > **Weakness 12: Discuss (or test) how occlusion, noise, or communication delay affects the learned geometry—important for practical multi-robot coordination.**
>
> **Response 12**:
>
> Thank you for your advice. In our shape-assembly task, occlusion is inherently addressed through the problem formulation. Our RL environment is partially observable, where each robot only perceives its 6 nearest neighbors. In dense shape assembly tasks, this setting functions as an implicit occlusion simulation where distant or occluded individuals are automatically filtered out. Our experimental results demonstrate that global geometric configurations emerge even under these constraints, **proving robustness to occlusion**.
>
>
> Regarding **communication delay**, since perception relies on relative positions, communication delays (which cause position drift) can be modeled as state observation noise over short timescales from a control theory perspective. Therefore, we indirectly verify tolerance to delay by evaluating the model's robustness to noise.
>
>
> Consequently, we conducted a **noise robustness analysis** to test agent performance under uncertainty. We injected Gaussian white noise ($\sigma = 0.02$) into the agents' observation vectors and retrained the policy. The results are presented in Appendix Figure 32. Panel (a) visualizes the assembly outcome, while panel (b) provides a quantitative analysis of the learned geometric structure. Despite increased value fluctuation caused by noise, internal critic values far exceed external ones. As the task progresses, the internal mean value decreases due to agent occupancy, yet the "In-Shape Top 10%" value remains significantly higher than that of the "Out-Shape" region.
>
> **These findings verify the robustness of the shape assembly task against various uncertainties.**

---

> ### Author Response · Authors · 2025-12-03
> **Response to Reviewer aMWe (Question 1 - 4)**
>
> > **Question 1: Can you provide statistics on how agents return to Voronoi boundaries after perturbation, compared with random baselines?**
>
> **Response 13**: Thank you for your suggestion. We compared our RL model against **three established baseline collective motion models** (Vicsek, Couzin, and Reynolds). These are classic models widely utilized in collective behavior research and offer better performance than purely random motion strategies. As shown in Figure 5(c), the RL model demonstrates superior stability compared to these baselines. For a detailed analysis, please refer to **Response 1**.
>
> > **Question 2:  How sensitive are ARM maps to probe orientation, initial position, or sampling resolution?**
>
> **Response 14**: Thank you for your question. In fact, the ARM successfully reveals the learned Voronoi diagram across these varying parameters, demonstrating its robustness to parameter changes. We refer the reviewer to **Response 4** for a detailed discussion.
>
> > **Question 3: How are Voronoi boundaries computed under periodic boundary conditions or multiple predators? Are they always continuous?**
>
> **Response 15**: Thanks for your question:  Actually, a Voronoi diagram partitions a plane into regions based on the distance to a set of points (predators). For multiple predators, the boundary is defined as the locus of points equidistant to the two nearest predators (Okabe et al., 2000).
>
> * **Continuity**: Mathematically, the distance function $d(x, p)$ in a Euclidean space is continuous. The Voronoi boundary is the level set where $d(x, p_i) - d(x, p_j) = 0$. Since the difference of two continuous functions is continuous, the zero-level set (the boundary) forms continuous line segments (edges) that meet at vertices. Therefore, Voronoi boundaries are always continuous within the domain.
>
> * **Computation under Periodic Boundary Conditions (PBC)**: Under PBC, the environment topology becomes a torus. The distance between a point and a predator is defined as the shortest geodesic distance on this torus. This is mathematically equivalent to tessellating the central simulation box with 8 copies of the environment, surrounding the actual domain. These 8 mirrored environments, together with the central one, form a $3 \times 3$ grid where predator motion is synchronized. We compute the Voronoi boundaries generated by the predators within this expanded superset and retain the segments falling within the central region to obtain the correct PBC boundaries.
>
> We provide our source code for computing Voronoi boundaries under periodic boundary conditions to facilitate reproducibility.
>
> > **Question 4: How many random seeds were used? What is the variance in emergent behavior across runs?**
>
> **Response 16**: We evaluated the system across 20 different random seeds. The results show that prey agents consistently evolve collective behaviors under RL training, with ARM successfully revealing the underlying Voronoi structure in all cases. An illustrative example is provided in Appendix **Figure 17**. Despite variations in the initial distribution of predators and prey, which alter the specific geometry of the Voronoi boundaries, the prey consistently converge toward these boundaries. This demonstrates that our method is robust and does not rely on specific seed initializations.

---

> ### Author Response · Authors · 2025-12-03
> **Response to Reviewer aMWe (Question 5 &6)**
>
> > **Question 5: How do results change with different prey/predator ratios or varying perceptual neighbor counts?**
>
> **Response 17**:
> Thank you for your question. In our original submission, we evaluated the impact of **varying prey/predator ratios** and **perceptual neighbor counts** on the RL-trained collective behaviors, as detailed in Appendix Figures 1 and 3. The results demonstrate that our collective model is robust to variations in perception and population size; specifically, collective behavior consistently emerges across different perception ranges, observed neighbor counts, and agent population sizes.
>
> To clarify this for the reader, we have revised the description of robustness in the main text:
>  "shown as Appendix A.2 confirms the robustness of emergent behaviors to alternative perception parameter settings."
>
> Furthermore, we have added ARM results for different prey/predator ratios in Appendix Figures 13–14. These results confirm that **ARM successfully identifies the underlying Voronoi structure even when agent populations change**. Consequently, both our RL model and the interpretability method effectively handle variations in agent quantity and perceptual parameters.
>
>
> > **Question 6: How does your learned strategy differ quantitatively from rule-based models (in clustering degree, survival rate, or energy efficiency)?**
>
> **Response 18**: Thanks for your question.
> We employed three metrics to evaluate the performance of different collective motion methods in pursuit-evasion scenarios:
> 1. **Steady-state Distance**: This measures the relative distance between prey agents and the Voronoi boundary after the system stabilizes.
> 2. **Deviation Distance**: This measures the displacement of prey agents following an impulsive perturbation.
> 3. **Survival Rate**: This is quantified by the number of capture collisions between prey and predators, where fewer collisions indicate a higher survival rate.
>
> The results for the first two metrics have been updated in Figure 5(c) of the revised manuscript. As detailed in Response 1, the RL model demonstrates superior stability compared to the baselines.
>
> Regarding collisions, we calculated the average number of collisions across 20 different random seeds.
> | Collective models | RL | Vicsek | Reynolds | Couzin |
> | :---: | :---: | :---: | :---: | :---: |
> | **Number of collisions** | 27 | 52 | 98 | 83 |
>
> The data indicates that the RL approach significantly reduces the frequency of predator captures, thereby enhancing the prey's survival rate.

---

### Author Response · Authors · 2025-12-02
**General Response to All Reviewers**

We sincerely appreciate the reviewers' time and constructive feedback. Beyond our specific responses to each reviewer, here we would like to highlight our contributions, the new experiments added during the rebuttal, and the major revisions made to the manuscript. The changes have been highlighted in blue in the revised paper attached to this submission.

### [**Our Contributions**]
We are glad to find out that the reviewers generally acknowledge the significance of our work:

1. **Fundamental and timely problem:** All reviewers agree that the paper addresses an important and underexplored question at the intersection of AI, biology, and complex systems [aMWe, PHNa, HkEp, NZCx].
2. **Novel methodology and strong originality:** The proposed interpretability framework, including the Agent Response Map (ARM), is a novel, intuitive, and valuable methodological contribution for analyzing agent policies [aMWe, PHNa, HkEp].
3. **Interesting and powerful mechanistic insights:** The discovery that agents implicitly learn geometric invariants (e.g., Voronoi boundaries as "line attractors") is highlighted as a powerful and interesting explanation for emergent coordination [aMWe, PHNa, HkEp].
4. **Clear presentation and structure:** The paper is considered readable and well-structured, with visualizations that effectively communicate the key findings [aMWe, PHNa].

### [**New Experiments**]
In this revision, we have added seven major experiments to address concerns regarding quantification, robustness, and generality:

1. **Expanded Quantitative Analysis:** We increased the quantitative metrics from one to three to validate the Voronoi Boundary claim.
    * **Attractor Dynamics:** Verified the robustness of the boundary as a line attractor by two different approaches: The first is manipulating predator positions to alter the Voronoi boundary geometry. The second is evaluating prey convergence under impulse perturbations against three baseline collective models (Vicsek et al., 1995; Reynolds, 1987; Couzin et al., 2002) [aMWe, PHNa, HkEp].
    * **Predictive validation:** Extended ARM from correlation analysis to causal prediction, quantifying how accurately Voronoi features predict agent aggregation zones [aMWe, PHNa].
    * **Sensitivity analysis:** We quantify changes in performance under variations in agent motion parameters [aMWe, PHNa, HkEp].

2. **Enhanced Sensitivity & Robustness Tests:**
    * **Parameter Sensitivity:** Beyond the original tests (orientation, velocity, seeds), we analyzed the impact of varying predator/prey population sizes, sampling resolutions, demonstrating ARM's robustness [aMWe, PHNa, HkEp].
    * **Environmental Diversity:** Validated the effectiveness of ARM in non-isotropic environments (e.g., with additional noise) and demonstrated computational scalability in high-dimensional (3D) spaces  [PHNa, HkEp].

3. **Generalizability to Cooperative Tasks:**
    * We expanded the multi-robot shape assembly task (a cooperative setting with global constraints) to include ARM analysis and quantitative validation [aMWe, PHNa, HkEp].
    * Tests were extended to complex non-convex and multi-component shapes under uncertainty (e.g., noise), confirming the framework's versatility [aMWe].

### [**Major Revisions to Main Text**]
We have implemented four primary changes to the main text:

1. **Completeness:** We moved the Related Work section and detailed RL training/network architecture specifications from the Appendix back to the Main Text [aMWe, NZCx].
2. **New Results (Figure 5):** We integrated three baseline comparisons and ARM discontinuity quantitative metrics directly into Figure 5, and revised the description of qualitative criteria in the Introduction to reflect these quantitative additions [aMWe, PHNa, HkEp].
3. **Expanded Application (Figure 8):** We added the ARM analysis and quantitative results for the shape assembly task to Figure 8, and rewrote the Abstract to highlight this extended application [aMWe, PHNa, HkEp].
4. **Clarifications & Refinements:**
    * Refined the description of agent perception [NZCx].
    * Provided comprehensive details for the ARM visualization in Figure 2 [NZCx].
    * Refined the description of causal feature ablation in the SHAP section [aMWe].
    * Elaborated on the parameter sensitivity analysis for ARM [aMWe, PHNa, HkEp].

### [**Major Revisions to Appendix**]
We have made four changes to the Appendix:

1. **Training Specifications:** Added detailed information regarding the optimizer, hardware infrastructure, and training duration [aMWe, NZCx].
2. **Sensitivity Analysis:** Analyzed the impact of varying predator/prey population sizes and ARM sampling resolutions [aMWe, PHNa, HkEp].
3. **Robustness Verification:** Provided additional ARM results across different random seeds [aMWe].
4. **Complex Scenarios:** Expanded the shape assembly experiments to test non-convex and multi-component shapes [aMWe, PHNa, HkEp].

---

### Meta-Review · Area_Chair_nU1i · 2026-01-08

**Summary:**

The paper proposes an explanatory framework, the Agent Response Map (ARM), to interpret how complex collective behaviors emerge from simple rewards in Multi-Agent Reinforcement Learning (MARL). The key finding suggests that agents implicitly learn geometric structures to coordinate their movements.

While the reviewers appreciate the motivation to demystify black-box MARL policies and find the visualization tools intuitive, the consensus is that the paper does not meet the bar for acceptance.

Several reviewers questioned whether the "Voronoi" finding is a significant discovery or merely a convoluted description of simple heuristic behaviors (e.g., maximizing distance from predators). The claim that these geometric structures act as "line attractors" lacks formal mathematical derivation or theoretical backing. The findings appear heavily dependent on specific, simple 2D environments, with insufficient evidence that the framework generalizes to complex or non-spatial tasks.

**Reviewer Concerns:**

Addressed:
- The authors successfully addressed Reviewer aMWe's request for quantitative metrics and comparisons with classical swarm models (Vicsek, Reynolds) during the rebuttal.
- The authors corrected the significant issue raised by Reviewer NZCx regarding the improper placement of core related work and implementation details in the appendix.
- Additional experiments were provided to demonstrate the robustness of ARM to noise and parameter variations, satisfying some concerns from Reviewer PHNa.

Outstanding:

- Reviewer NZCx and others remain unconvinced that the proposed explanation offers true insight beyond "agents learn to run away." The rebuttal failed to prove that the "Voronoi" mechanism is a distinct, learned strategy rather than a trivial optimal solution to the reward function.
- Reviewer HkEp's concern regarding the lack of theoretical support for the "attractor" property remains unresolved. The provided empirical Monte Carlo verification was deemed insufficient to substitute for mathematical proof.
- Reviewers PHNa and HkEp maintain that the contribution is primarily descriptive/interpretive and restricted to simple setups, lacking algorithmic contributions that would improve performance or generalization to complex terrains.

**Reviewer Scores:**

* **Reviewer aMWe:** Initial Score: 6
    * **Estimated Final Score: 8.** The authors provided extensive rebuttal to this reviewer. Questions are well-addressed.
* **Reviewer PHNa:** Initial Score: 4
    * **Estimated Final Score: 4.** The reviewer's fundamental concern about the work being "limited generality" and "primarily interpretive" was not changed by the rebuttal experiments.
* **Reviewer HkEp:** Initial Score: 4
    * **Estimated Final Score: 4.** The lack of theoretical derivation for the core "attractor" claim remains a dealbreaker for this reviewer.
* **Reviewer NZCx:** Initial Score: 2
    * **Estimated Final Score: 2.** While the authors fixed the formatting/ethics issue, the reviewer's harsh critique regarding the lack of technical novelty remains outstanding.

---

### Decision · Program_Chairs · 2026-01-26

Reject